# LoKO: Low-Rank Kalman Optimizer for Online Fine-Tuning of Large Models

## Abstract

Training large models with millions or even billions of parameters from scratch incurs substantial computational costs. Parameter Efficient Fine-Tuning (PEFT) methods, particularly Low-Rank Adaptation (LoRA), address this challenge by adapting only a reduced number of parameters to specific tasks with gradient-based optimizers. In this paper, we cast PEFT as an optimal filtering/state estimation problem and present **Lo**w-Rank **K**alman **O**ptimizer (**LoKO**) to estimate the optimal trainable parameters in an online manner. We leverage the low-rank decomposition in LoRA to significantly reduce matrix sizes in Kalman iterations and further capitalize on a diagonal approximation of the covariance matrix to effectively decrease computational complexity from quadratic to linear in the number of trainable parameters. Moreover, we discovered that the initialization of the covariance matrix within the Kalman algorithm and the accurate estimation of the observation noise covariance are the keys in this formulation, and we propose robust approaches that work well across a vast range of well-established computer vision and language models. Our results show that LoKO converges with fewer iterations and yields better performance models compared to commonly used optimizers with LoRA in both image classifications and language tasks. Our study opens up the possibility of leveraging the Kalman filter as an effective optimizer for the online fine-tuning of large models.

## 1 Introduction

The widespread adoption of deep neural networks, particularly large models, across various fields is mainly driven by the pre-training on extensive datasets followed by task-specific fine-tuning (Han et al., 2024). In recent years, the concept of online fine-tuning has attracted significant attention across diverse domains, ranging from robotics (Yang et al., 2024; Fang et al., 2022; Julian et al., 2020), reinforcement learning (Zheng et al., 2022; Nakamoto et al., 2024), computer vision (Gao et al., 2023; Kang et al., 2020), and natural language processing (Fan et al., 2024). Online fine-tuning refers to the process of continually updating a pre-trained model's parameters as new data in a temporal stream of data becomes available, typically during deployment or in real-time applications. As online full fine-tuning requires substantial computational resources and can negatively impact the generalization capabilities(Han et al., 2024), Parameter-Efficient Fine-Tuning (PEFT) techniques freeze the majority of the model parameters and selectively update a smaller subset. Among PEFT approaches, the Low-Rank Adaptation (LoRA) technique has recently been widely recognized due to its efficient adaptation with low computational overhead (Han et al., 2024). Many extensions to LoRA have also been proposed to improve its learning capacity and training stability (Liu et al., 2024; Zhang et al., 2023; Renduchintala et al., 2023; Dettmers et al., 2024; Valipour et al., 2022).

In these PEFT approaches, stochastic gradient descent (SGD) has been the dominant method for the parameter optimization. The adaptive first-order optimizers, such as Adam (Kingma & Ba, 2014) and its variants, have demonstrated superior performance compared to traditional SGD ones(Ruder, 2016), as evidenced by their widespread adoption in LoRA extensions. Despite the simplicity and efficacy of these gradient-based optimizers, they exclusively rely on first-order derivatives, which may result in (sub-)optimal convergence and inefficient optimization (Reddi et al., 2019).

In contrast, many previous works (Singhal & Wu, 1988; 1989; Puskorius & Feldkamp, 1991; Williams, 1992; Heimes, 1998; Rivals & Personnaz, 1998) showed that recursive Extended Kalman

Filter (EKF) algorithm – a method for state estimation of nonlinear systems using a data stream introduced by Kalman & Bucy (1961) – can optimize relatively small models with performance surpassing the gradient-based counterparts. This achievement remained obscure until when Ollivier (2018) theoretically demonstrated that EKF is effectively equivalent to online natural gradient descent. This offers a new perspective on advanced optimization: instead of relying on complex techniques, we can recursively infer the optimal parameters as state estimation. However, despite the commendable performance of the EKF in training neural models, its practicality has been hampered, especially with the advent of large models. Pariticularly, the EKF algorithm involves several sequential operations, including Linearization, Prediction, and Update steps, all of which entail significant computational overhead. The size of the crucial covariance matrix in EKF can even grow quadratically with the number of model parameters involved in training.

In this paper, we leverage the low-rank decomposition technique from LoRA to reduce the number of trainable parameters in specific layers. We show that the EKF algorithm can be particularly useful for fine-tuning as fine-tuning only involves a small portion of model parameters. We introduce LoKO, a Kalman-based training algorithm, as an alternative to advanced optimizers for online fine-tuning of large models. Particularly, our contributions are:

- Based on the Low-Rank Adaptation (LoRA) method, introduced by Hu et al. (2021), which significantly reduces the number of trainable parameters, we demonstrated its compatibility with the Kalman filter. Also, we showed that this combination offers faster performance than traditional optimizers in online fine-tuning scenarios.

- We employ a diagonal approximation for the covariance matrix $P$, a common approach to reduce the computational overhead from quadratic to linear. By integrating this with the exponential moving average (EMA) for estimating the matrix $R$ and incorporating it into the LoRA framework, we achieve improved performance without additional computational cost.

- We conduct various experiments to demonstrate LoKO's performance in online fine-tuning classification tasks across computer vision and language modeling domains.

- To the best of our knowledge, this is the first successful attempt to fine-tune large and complex models, including transformers with millions of parameters, using the Kalman filter algorithm.

In summary, LoKO shows outstanding performance on computer vision and language modeling benchmarks: MNIST (LeCun et al., 1998), CIFAR-10/100 (Krizhevsky et al., 2009), ImageNet100 (Vinyals et al., 2016), SST-2 (Socher et al., 2013), COLA (Warstadt, 2019), MRPC (Dolan & Brockett, 2005). This paper contributes to ongoing efforts to develop a more efficient and fast optimization algorithm for online fine-tuning of increasingly more complex large models.

## 2 RELATED WORK

**Parameter-Efficient Fine-Tuning (PEFT):** Traditional full fine-tuning typically demands significant computational resources and may damage the model's generalization ability, occasionally leading to catastrophic forgetting (Han et al., 2024). In contrast, Parameter-Efficient Fine-Tuning (PEFT) efficiently freezes a portion of the parameters while updating a reduced number of trainable ones to mitigate these issues. Three primary approaches for PEFT are commonly used: plug-in adapters, parameter freezing and model reparameterization. Plug-in adapters refer to the techniques of introducing an extra trainable adapter module to the pre-trained model, as demonstrated in works such as (Chen et al., 2024; Gao et al., 2024). Parameter freezing is to freeze selected model parameters and update only a targeted subset, like BitFit (Zaken et al., 2021), Child-Tuning (Xu et al., 2021), IA$^3$ (Liu et al., 2022), and FISH Mask (Sung et al., 2021). Model reparameterization, as the name suggests, reparameterizes model parameters using typically the Low-Rank Adaptation (LoRA) technique, which adds low-rank weight matrices as trainable parameters(Hu et al., 2021). Multiple **extensions to LoRA** have been proposed to improve learning capacity and training stability. For instance, DoRA (Liu et al., 2024) decomposes pre-trained weights into magnitude and direction components to minimize the number of trainable parameters more efficiently. AdaLoRA(Zhang

et al., 2023) utilizes singular value decomposition (SVD) to dynamically allocate the parameter budget based on importance scoring. Tied-LoRA(Renduchintala et al., 2023) leverages weight tying and selective training to reduce the number of trainable parameters further. To optimize the memory efficiency, QLoRA (Dettmers et al., 2024), QA-LoRA (Xu et al., 2023), and LoftQ (Li et al., 2023) address the issue of memory usage by quantization technique. DyLoRA (Valipour et al., 2022) is a dynamic search-free LoRA to avoid exhaustive search for the most optimal rank. However, all these extensions utilize the Adam optimizer or its variants like AdamW in parameter optimization.

**Kalman Filter for Optimizing Neural Networks:** The idea of using the Kalman filter as parameter optimization in deep learning comes from Singhal & Wu (1988), which showed that the process of training neural networks can be conceptualized as tackling a system identification challenge for a nonlinear dynamic system, and thus Extended Kalman Filter (EKF) can be used to train neural network parameters. The superior performance of the Kalman-based training algorithm compared to the traditional backpropagation technique drew attention to exploring the relationship between these two classical algorithms (Ruck et al., 1992; Ollivier, 2018). To make the Kalman filter more scalable, several studies have addressed the computational complexity associated with this training algorithm, notably using matrix partitioning techniques (Shah & Palmieri, 1990; Shah et al., 1992; Puskorius & Feldkamp, 1991), and a low-dimensional (block-)diagonal approximation of the covariance matrix (Murtuza & Chorian, 1994). Only recently, Ollivier (2018; 2019) demonstrated that training with a Kalman filter is, in fact, equivalent to an online natural gradient descent. This finding renewed interest in this training method once again. Various studies have since addressed the scalability challenges of the EKF optimizer. For example, Chang et al. (2022) introduced a diagonal Gaussian approximation, while Chang et al. (2023) proposed a low-rank plus diagonal decomposition of the posterior precision matrix. Hennig et al. (2024) developed a matrix-free iterative algorithm to further enhance efficiency. In the context of factorization models, Gómez-Uribe & Karrer (2021) introduced a decoupled EKF (DEKF). Furthermore, EKF has been applied to several specialized areas, such as continual learning (Titsias et al., 2023), test-time adaptation (Schirmer et al., 2024), and reinforcement learning (Shashua & Mannor, 2019; 2020; Shashua & Mannor; Totaro & Jonsson, 2021; Shih & Liang, 2024). Other notable work includes loss adaptivity in the Kalman optimization algorithm (Davtyan et al., 2022), the Bayesian online natural gradient (Jones et al., 2024), and approaches for handling nonstationary data in online learning (Jones et al., 2022b;a). However, the practical implementation of these methods remained infeasible for large models.

## 3 PRELIMINARIES AND BACKGROUND

### 3.1 EXTENDED KALMAN FILTER (EKF)

Consider a general state-space model:

$$\boldsymbol{s}_k = f(\boldsymbol{x}_k, \boldsymbol{s}_{k-1}) + \boldsymbol{w}_k, \tag{1a}$$

$$\boldsymbol{y}_k = h(\boldsymbol{x}_k, \boldsymbol{s}_k) + \boldsymbol{v}_k, \tag{1b}$$

where $\boldsymbol{s}_k \in \mathbb{R}^n$, $\boldsymbol{y}_k \in \mathbb{R}^m$, and $\boldsymbol{x}_k \in \mathbb{R}^p$ denote the states, measurement, and input vectors at time $k$, respectively. In this framework, the function $f(\cdot)$ is called the process model or the state transition function, and $h(\cdot)$ is the measurement model or observation function. In addition, $\boldsymbol{w}_k \sim \mathcal{N}(0, \boldsymbol{Q}_k)$ and $\boldsymbol{v}_k \sim \mathcal{N}(0, \boldsymbol{R}_k)$ represent process noise and observation noise, respectively. These noises are assumed to have known distributions, typically white Gaussian noise with zero mean. The problem of state estimation for a nonlinear system, as depicted by equation 1a, can be addressed through the well-established recursive EKF algorithm (Welch et al., 1995). In the following, we detail the various steps involved in implementing the extended Kalman filter (EKF):

- **Prediction:**

$$\boldsymbol{s}_{k|k-1} = f(\boldsymbol{x}_k, \boldsymbol{s}_{k-1}) \tag{2a}$$

$$\boldsymbol{P}_{k|k-1} = \boldsymbol{F}_k \boldsymbol{P}_{k-1} \boldsymbol{F}_k^\top + \boldsymbol{Q}_k \tag{2b}$$

where $\boldsymbol{s}_{k|k-1}$ and $\boldsymbol{P}_{k|k-1}$ are predicted (or *prior*) states and covariance, respectively. $\boldsymbol{F}_k$ denotes the Jacobian matrix of the function $f(\cdot)$ with respect to states at time $k$.

- **Updating:**

$$\boldsymbol{K}_k = \boldsymbol{P}_{k|k-1} \boldsymbol{H}_k^\top (\boldsymbol{H}_k \boldsymbol{P}_{k|k-1} \boldsymbol{H}_k^\top + \boldsymbol{R}_k)^{-1} \tag{3a}$$

$$\boldsymbol{s}_k = \boldsymbol{s}_{k|k-1} + \boldsymbol{K}_k (\boldsymbol{y}_k - h(\boldsymbol{x}_k, \boldsymbol{s}_{k|k-1})) \tag{3b}$$

$$\boldsymbol{P}_k = \boldsymbol{P}_{k|k-1} - \boldsymbol{K}_k \boldsymbol{H}_k \boldsymbol{P}_{k|k-1} \tag{3c}$$

where $\boldsymbol{K}_k$ is the Kalman gain, and $\boldsymbol{H}_k$ indicates the Jacobian matrix of the function $h(\cdot)$ with respect to the states at time $k$. Finally, $\boldsymbol{s}_k$ and $\boldsymbol{P}_k$ are updated (or *posterior*) states and covariance matrix.

## 3.2 Low-Rank Adaptation (LoRA)

Low-Rank Adaptation (LoRA) (Hu et al., 2021) is a technique designed to efficiently fine-tune large pre-trained neural networks by reducing the number of trainable parameters. Instead of updating the entire pre-trained weight matrix, LoRA introduces a low-rank decomposition that captures the essential changes needed for fine-tuning. Consider a layer in a neural network with a pre-trained weight matrix $\boldsymbol{W}_0 \in \mathbb{R}^{d \times q}$, where $d$ and $q$ represent the dimensions of the weight matrix. The output of this layer can be expressed as:

$$\boldsymbol{z} = \boldsymbol{W}_0 \boldsymbol{x} + \Delta \boldsymbol{W} \boldsymbol{x} \tag{4}$$

Here, $\boldsymbol{x}$ is the input vector, and $\Delta \boldsymbol{W}$ represents the adjustment to the weights during fine-tuning. LoRA modifies this by introducing two smaller matrices, $\boldsymbol{A} \in \mathbb{R}^{r \times q}$ and $\boldsymbol{B} \in \mathbb{R}^{d \times r}$, where $r$ is the chosen rank with $r \ll \min(d, q)$. The update to the weight matrix is then expressed as:

$$\boldsymbol{z} = \boldsymbol{W}_0 \boldsymbol{x} + \boldsymbol{B} \boldsymbol{A} \boldsymbol{x} \tag{5}$$

At the beginning of training, the matrix $\boldsymbol{A}$ is initialized with a random Gaussian distribution $\mathcal{N}(\boldsymbol{0}, \sigma^2 \boldsymbol{I})$, and matrix $\boldsymbol{B}$ is initialized to zero. Using this low-rank decomposition, LoRA reduces the number of trainable parameters from $d \times q$ to $r \times (d + q)$, where $r$ is much smaller than $d$ and $q$. This technique can be applied on certain layers of a neural network model, $h(\boldsymbol{x}, \boldsymbol{\theta})$, resulting in a re-parameterized version, $h_{LoRA}(\boldsymbol{x}, \tilde{\boldsymbol{\theta}})$, with a reduced number of trainable parameters. This reduction enables efficient fine-tuning of large models, making it feasible to apply the Kalman filter.

## 4 Low-rank Kalman Optimizer

### 4.1 Kalman Formulation for LoRA

Consider a pre-trained model in which certain layers are scaled down using the LoRA method. The modified model is parameterized by a reduced set of trainable parameters $\tilde{\boldsymbol{\theta}}_k \in \mathbb{R}^{\tilde{n}}$, where $\tilde{n} \ll n$:

$$\hat{\boldsymbol{y}}_k = h_{LoRA}(\boldsymbol{x}_k, \tilde{\boldsymbol{\theta}}_k). \tag{6}$$

Here, $\boldsymbol{x}_k \in \mathbb{R}^p$ denotes the model input and $\hat{\boldsymbol{y}}_k \in \mathbb{R}^m$ represents the predicted output in the $k^{th}$ observed data. Let us adopt $\boldsymbol{y}_k$ as the true output. The $\hat{\boldsymbol{y}}_k$ represents the predicted values for the regression tasks and the predicted probabilities for the classification tasks. In both scenarios, the predicted output $\hat{\boldsymbol{y}}_k$ can be interpreted as the mean parameter of a Gaussian distribution over the actual output $\boldsymbol{y}_k$. This relationship can be expressed as white noise as $\boldsymbol{v}_k \sim \mathcal{N}(\boldsymbol{0}, \boldsymbol{R}_k)$, where $\boldsymbol{R}_k = \mathrm{Cov}(\boldsymbol{y}_k | \hat{\boldsymbol{y}}_k)$. More broadly, $\hat{\boldsymbol{y}}_k$ serves as the mean parameter for an exponential family distribution over $T(\boldsymbol{y}_k)$, where $T(\cdot)$ denotes the sufficient statistics for the exponential family. In this case, $\boldsymbol{R}_k = \mathrm{Cov}(T(\boldsymbol{y}_k) | \hat{\boldsymbol{y}}_k)$ denotes the covariance matrix of the exponential family distribution.

Several approximations for the matrix $\boldsymbol{R}_k$ have been proposed in the literature. One common approach is to approximate it as the identity matrix, $\boldsymbol{R}_k \approx \boldsymbol{I}$, as shown in (Puskorius & Feldkamp, 1991; Murtuza & Chorian, 1994). Other formulations include $\boldsymbol{R}_k = \boldsymbol{I} \cdot e^{-k/50}$(Singhal & Wu, 1988; 1989), and a more recent approximation, $\boldsymbol{R}_k = \mathrm{diag}(\hat{\boldsymbol{y}}_k) - \hat{\boldsymbol{y}}_k \hat{\boldsymbol{y}}_k^\top$(Ollivier, 2018; Chang et al., 2022). To obtain a more precise approximation of $\boldsymbol{R}_k$, we employ an Exponential Moving Average (EMA) approach based on the definition of the covariance matrix for estimating the matrix $\boldsymbol{R}_k$. We make the simplifying assumption that $\tilde{\boldsymbol{\theta}}_k \approx \tilde{\boldsymbol{\theta}}_{k|k-1}$, allowing us to compute the covariance

matrix as follows:

$$\boldsymbol{R}_k = \beta \boldsymbol{R}_{k-1} + (1 - \beta)\hat{\boldsymbol{R}}_k, \tag{7a}$$

$$\text{where} \quad \hat{\boldsymbol{R}}_k = \left(\boldsymbol{y}_k - h_{LoRA}(\boldsymbol{x}_k, \tilde{\boldsymbol{\theta}}_{k|k-1})\right)\left(\boldsymbol{y}_k - h_{LoRA}(\boldsymbol{x}_k, \tilde{\boldsymbol{\theta}}_{k|k-1})\right)^{\top}, \tag{7b}$$

and $\beta \in (0, 1)$ is the forgetting factor. The proofs and detailed derivations can be found in Appendix B for more details.

LoKO estimates the (sub-)optimal values of LoRA matrices $\boldsymbol{A}$ and $\boldsymbol{B}$ in real-time data streams through the Kalman algorithm. To formulate the online fine-tuning problem within the Kalman filtering framework, we assume there are no feedback loops in machine learning model, for example, a feed-forward neural network and transformers. This assumption enables us to consider the trainable parameters that are represented as the state vector of the process model ($\tilde{\boldsymbol{\theta}}_k \equiv \boldsymbol{s}_k$). Furthermore, the process model (or transition function) can be modeled as an identity function $f(\boldsymbol{x}_k, \tilde{\boldsymbol{\theta}}_{k-1}) = \tilde{\boldsymbol{\theta}}_{k-1}$ with no process noise ($\boldsymbol{w}_k = 0$), which provides the prediction of the states at the next time step as $\tilde{\boldsymbol{\theta}}_k = \tilde{\boldsymbol{\theta}}_{k-1}$. Finally, by defining $\boldsymbol{y}_k = h_{LoRA}(\boldsymbol{x}_k, \tilde{\boldsymbol{\theta}}_k) + \boldsymbol{v}_k$ as the measurement model (or observation function), we can apply the recursive Kalman filter algorithm to estimate (sub-)optimal values of $\tilde{\boldsymbol{\theta}}_k$.

Although the low-rank decomposition by LoRA offers a significant reduction in parameter size compared to the original parameter space $n$, the size of the covariance matrix $\boldsymbol{P}$ scales quadratically with the number of trainable parameters $\tilde{n}$. For large models such as deep neural networks, characterized by high-dimensional trainable parameters, the computational cost of implementing the Kalman algorithm becomes prohibitively expensive due to its $\tilde{n}^2$ complexity. To address this challenge, one strategy involves decoupling the update phase of the Kalman filtering algorithm into smaller partitions, as outlined by Puskorius & Feldkamp (1991), which, however, may be infeasible for large models with millions of trainable parameters. The other approach is to approximate the covariance matrix $\boldsymbol{P}$ with low-dimensional matrices. Our empirical findings demonstrate that as the fine-tuning algorithm progresses, the covariance matrix of the feed-forward neural network asymptotically approaches a (block-)diagonal configuration:

$$\mathbb{E}[\hat{\boldsymbol{p}}] = \text{diag}(\boldsymbol{P}). \tag{8}$$

Therefore, we adopt a diagonal approximation of the covariance matrix, denoted as $\hat{\boldsymbol{p}}$. This approximation significantly reduces both computational and storage costs.

**Proposition 1.** *Leveraging the low-rank decomposition technique in LoRA and applying the diagonal approximation of covariance matrix, the steps of the Low-Rank Kalman Optimizer (LoKO) can be outlined below:*

- *Prediction:*

$$\tilde{\boldsymbol{\theta}}_{k|k-1} = \tilde{\boldsymbol{\theta}}_{k-1} \tag{9a}$$

$$\hat{\boldsymbol{p}}_{k|k-1} = \hat{\boldsymbol{p}}_{k-1} \tag{9b}$$

- *Pre-Updating:*

$$\boldsymbol{R}_k = \beta \boldsymbol{R}_{k-1} + (1 - \beta)\hat{\boldsymbol{R}}_k \tag{10a}$$

- *Updating:*

$$\boldsymbol{K}_k = \hat{\boldsymbol{p}}_{k|k-1} \bullet \boldsymbol{H}_k^{\top} \left(\boldsymbol{H}_k(\hat{\boldsymbol{p}}_{k|k-1} \bullet \boldsymbol{H}_k^{\top}) + \boldsymbol{R}_k\right)^{-1} \tag{11a}$$

$$\tilde{\boldsymbol{\theta}}_k = \tilde{\boldsymbol{\theta}}_{k|k-1} + \boldsymbol{K}_k(\boldsymbol{y}_k - h_{LoRA}(\boldsymbol{x}_k, \tilde{\boldsymbol{\theta}}_{k|k-1})) \tag{11b}$$

$$(\hat{\boldsymbol{p}}_k)^i = \left(\hat{\boldsymbol{p}}_{k|k-1}\right)^i - \left(\boldsymbol{K}_k\right)_j^i \left(\boldsymbol{H}_k\right)_i^j \left(\hat{\boldsymbol{p}}_{k|k-1}\right)^i \tag{11c}$$

*where the symbol $\bullet$ represents the transposed Khatri–Rao product, which is essentially the row-by-row Kronecker product of the vector $\hat{\boldsymbol{p}}_{k|k-1}$ and matrix $\boldsymbol{H}_k^{\top}$. The equation 11c represents the diagonal update of the covariance matrix, expressed with Einstein notation. For more details, see Appendix B.*

Note that the operations used in equation 11a, and equation 11c can be computed efficiently. For example, in PyTorch, they can be seamlessly implemented using the $*$, and **einsum**() operators, streamlining the computational process.

Importantly, in the above formulation, the matrix inversion required in the Kalman gain equation 11a has a dimension of $m^2$, where $m$ represents the size of the model output. This implies that the computational cost of the matrix inversion remains constant regardless of the size of the model. Consequently, whether the model is small or large, the computational expense of this operation remains constant, offering a consistent performance characteristic across different model sizes. In contrast, many advanced optimizers such as the Natural Gradient Descent (NGD) necessitate the inversion of the full pre-condition matrix (like the Fisher information matrix in NGD), which is of high dimensionality. Moreover, the computation of the Jacobian matrix $\boldsymbol{H}_k$ does not require individual backpropagation processes for each output component. Leveraging GPU capabilities allows for parallel computation and thus efficiently streamlines the process.

### 4.2 INITIALIZATION OF **P** & APPROXIMATION OF **R**

**Initialization of $\hat{\boldsymbol{p}}_0$:** Our findings demonstrate that the initialization of $\hat{\boldsymbol{p}}_0$ plays a crucial role in the performance of the filter as a training algorithm. High values of $\hat{\boldsymbol{p}}_0$ indicate high uncertainty or lack of confidence in the initial learning parameters, which can result in the Kalman filter making large corrections and experiencing potential instability and divergence. In contrast, initialization $\hat{\boldsymbol{p}}_0$ with very low values suggests high confidence in the initial learning parameters. In this case, the filter will heavily trust the initial model parameters. If these initial parameters are inaccurate, this can lead to slow updates or even no updates at all, as the filter gives insufficient weight to new measurements. Proper initialization of $\hat{\boldsymbol{p}}_0$ balances these extremes, ensuring that the filter can adapt appropriately to new data while maintaining stability and accuracy throughout the training process. Therefore, it is essential to establish both upper and lower bounds for $\hat{\boldsymbol{p}}_0$. To ensure the initialization of $\hat{\boldsymbol{p}}_0 = [p_1, p_2, ... p_i, ..., p_n]$ yields a positive definite diagonal matrix, we examined two methods for initialization of $\hat{\boldsymbol{p}}_0$:

- **Method 1**: Setting a constant positive value: $p_i = c \quad \forall i$.
- **Method 2**: Assigning random positive values drawn from a uniform distribution: $p_i \sim U(0, \text{upper\_bound}) \quad \forall i$.

Our experiments show that precisely estimating the $\boldsymbol{R}_k$ matrix greatly influences the bounds of $\hat{\boldsymbol{p}}_0$. The more accurate the estimation of $\boldsymbol{R}_k$, the broader the acceptable range for the initial $\hat{\boldsymbol{p}}_0$.

**Approximation of $\boldsymbol{R}_k$:** In addition to the method described in equation 7, we propose an alternative method for approximating the observation noise covariance, $\boldsymbol{R}_k$, with enhanced accuracy and without additional computation cost. Specifically, we incorporate an additional term from the first-order Taylor to approximate changes more precisely:

$$\boldsymbol{R}_k = \beta \boldsymbol{R}_{k-1} + (1 - \beta)\hat{\boldsymbol{R}}_k, \tag{12a}$$

$$\text{where} \quad \hat{\boldsymbol{R}}_k = \left(\boldsymbol{y}_k - h_{LoRA}(\boldsymbol{x}_k, \tilde{\boldsymbol{\theta}}_{k|k-1})\right)\left(\boldsymbol{y}_k - h_{LoRA}(\boldsymbol{x}_k, \tilde{\boldsymbol{\theta}}_{k|k-1})\right)^\top + \boldsymbol{H}_k(\hat{\boldsymbol{p}}_{k|k-1} \bullet \boldsymbol{H}_k^\top) \tag{12b}$$

with the forgetting factor of $\beta \in (0, 1)$. Note that this method will not add extra computational cost since the operation of $\boldsymbol{H}_k(\hat{\boldsymbol{p}}_{k|k-1} \bullet \boldsymbol{H}_k^\top)$ will be part of the Kalman gain calculation in equation 11a. See Appendix B for more details.

## 5 EXPERIMENTS AND ANALYSIS

### 5.1 EXPERIMENTS SETUP

We assess the performance of LoKO by implementing it in various well-established computer vision and language models. Our computer vision experiments involve online fine-tuning for image classification on the MNIST dataset using DenseNet-121 with 7 million parameters (Huang et al., 2017),

the CIFAR-10/100 dataset with ResNet-18 and ResNet-50 (He et al., 2016), and ViT-B16 (Dosovitskiy, 2020), which contain 12 million, 26 million, and 86 million parameters, respectively. Furthermore, we evaluated ImageNet100 using ViT-L16 (Dosovitskiy, 2020), 305 million parameters. For text classification tasks, we examine the SST-2, CoLA, and MRPC datasets using RoBERTa-base and RoBERTa-large (Liu, 2019), which have 125 million and 355 million parameters, respectively. We employ the following pre-trained models as backbones for our fine-tuning experiments:

- For MNIST/DenseNet-121, we used a backbone pre-trained on ImageNet via PyTorch Image Models (timm) (Wightman et al., 2023).
- For CIFAR-10/ResNet18 and CIFAR-100/ResNet50, we employed pre-trained models from (He et al., 2016), which were trained on ImageNet as the backbone.
- For CIFAR-10/ViT-B16, CIFAR-100/ViT-B16, and ImageNet100/ViT-L16, we used DINOv2 (Oquab et al., 2023), which was pre-trained on ImageNet.
- For the language tasks we employed RoBERTa pre-trained model from (Liu, 2019).

We use AdamW (Loshchilov & Hutter, 2017) and AdaGrad (Duchi et al., 2011) as baseline methods since they are widely used in LoRA. We set the batch size to 1 so that the learning algorithms train the models in the context of online classification tasks at each time step. We configure AdamW with a linear learning rate decay schedule (weight_decay=0.0001), starting from 1e-4, which empirically yields the highest accuracy after a preliminary sweep of various learning rates and schedules. In addition, $\beta_1$ and $\beta_2$ for this optimizer are set to 0.9 and 0.999, respectively. Furthermore, the learning rate of AdaGrad is set to 0.001 with no weight decay. The hyperparameters in our LoKO algorithm are set according to our discussion in 5.3. The diagonally approximated covariance vector $\hat{\boldsymbol{p}}_k$ is initialized with random numbers sampled from a uniform distribution between 0 and 0.2. Furthermore, we use our second method for $\hat{\boldsymbol{R}}_k$ estimation, and the forgetting factor for the EMA is set to $\beta = 0.95$. We also adopt the pre-trained models introduced by (Caron et al., 2021) as the backbone for our fine-tuning experiments with ViT. The low-rank decomposition is applied on *query* layers in transformer-based architectures and *convolution* layers in CNN networks. All experiments are conducted on an A100 GPU platform three times and the average and standard deviation of them are reported.

## 5.2 MAIN RESULTS

We use two primary evaluation metrics in our study. First, we calculate the moving average of the *loss* with a window size of 1000. Second, we measure the *average online accuracy*, as adopted by Cai et al. (2021), up to the current timestep $k$. The average online accuracy is defined as $\text{acc}(k) = \frac{1}{k} \sum_{i=1}^{k} \mathbb{1}_A(\boldsymbol{y}_i)$, where $\mathbb{1}_{\{\cdot\}}(\cdot)$ is an indicator function, and $A$ is the set of Top1 or Top5 prediction. This metric is computed online during the training process and serves to evaluate the algorithm's capacity to incorporate new knowledge in real-time. We evaluate the convergence speed of LoKO by comparing the number of observations required to achieve equivalent loss or accuracy levels across different algorithms. Figure 1, 2 illustrate LoKO's performance in online fine-tuning for image and language classification, compared to LoRA with AdamW and AdaGrad, across various well-established models and datasets. The figures illustrate L1 losses for images and cross-entropy losses for texts, along with accuracy during training. These results demonstrate that LoKO consistently matches or surpasses the performance of alternative optimizers. Additionally, Tables 1 and 2 present the average and standard deviation of online accuracy for a single epoch of online fine-tuning. Note that for COLA and MRPC datasets, we performed multiple epochs due to their smaller dataset sizes. These results demonstrate that across all models and datasets, LoKO consistently outperforms (and in some cases performs almost equivalently) the other methods by achieving the highest average online accuracy with LoKO. The lower standard deviation in LoKO's results also highlights its more consistent and robust performance compared to other methods. Furthermore, it is important to emphasize that (sub-)optimal values of the trainable parameters can be attained across a range of LoKO hyperparameters, provided they fall within the permissible initialization interval. This contrasts with gradient-based optimizers, whose performance is highly sensitive to the selection of appropriate hyperparameters, particularly the learning rate.

Finally, as can be seen in Table 2, LoKO demonstrates slightly lower performance than AdamW on certain language tasks. A justification for this observation is the fact that the AdamW optimizer benefits from the decoupled weight decay, which has contributed to its strong performance in language modeling tasks (Xie & Li, 2024). On the other hand, the gradient-free Kalman algorithm used

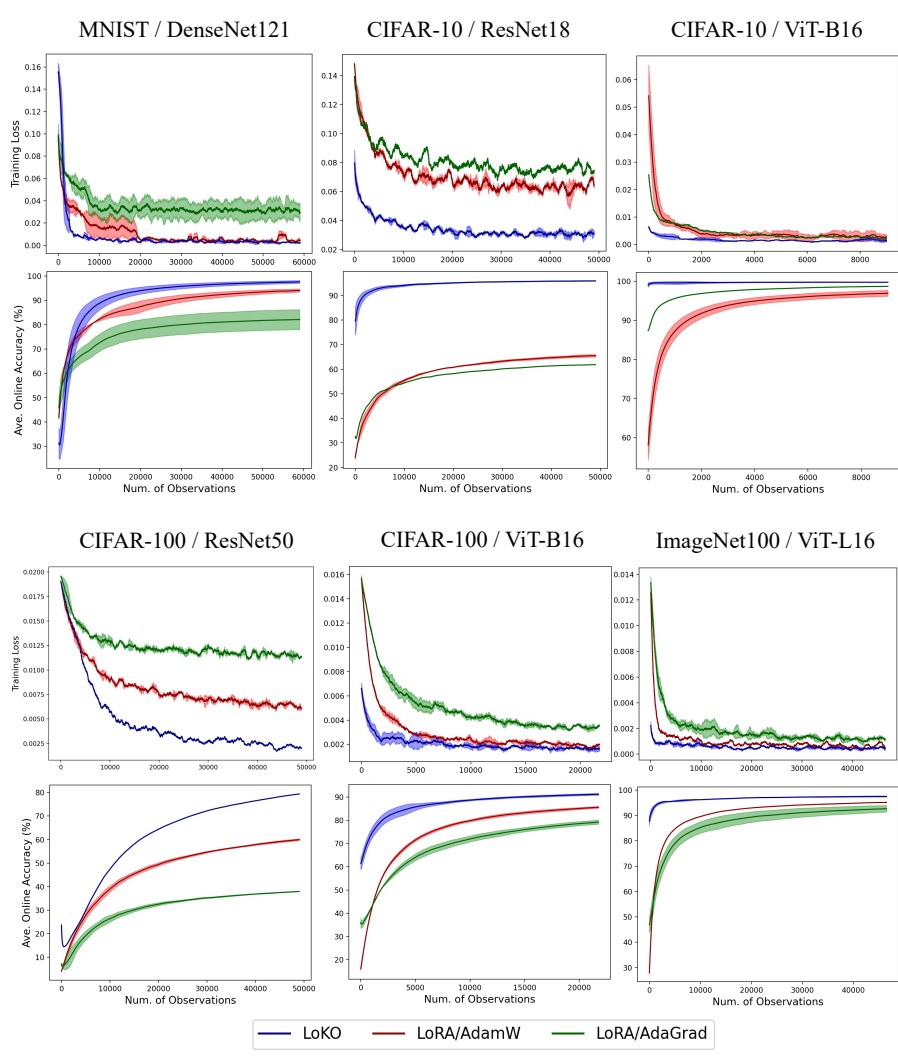

Figure 1: Performance of LoKO (blue) compared to LoRA/AdamW (red) and LoRA/AdaGrad (green) for different computer vision datasets and models. The upper rows show the training loss, and the lower rows display the average online accuracy versus the number of observed data.

in LoKO does not incorporate a similar regularization mechanism, which may explain the slightly lower performance in certain cases compared to AdamW.

At the end, we conducted supplementary experiments by applying the Kalman filter on the DoRA variant of LoRA. These experiments were performed on transformer-based models, and the results were compared against those obtained using AdamW and AdaGrad. The corresponding outcomes are presented in Figure 3 and 4 in Appendix D as well as Table 1 and 2. As shown, the results are promising for the Kalman algorithm, demonstrating its compatibility and effectiveness when integrated with this Weight-Decomposed Low-Rank Adaptation technique (Liu et al., 2024).

### 5.3 ABLATION OF INITIALIZATION AND COVARIANCE APPROXIMATION

**Initialization of $\hat{p}_0$:** Initializing $\hat{p}_0$ requires prior knowledge of the network weights and their associated uncertainties. In the absence of such knowledge, almost any initial values for $\hat{p}_0$ can be chosen, provided they are not too close to zero, and in some cases, they are not too far from an upper bound. To develop a comprehensive understanding of the behavior of the initialization of $\hat{p}_0$, we begin by evaluating its impact on training from scratch using the Kalman filter. The MNIST

Table 1: Results for MNIST, CIFAR-10/100, and ImageNet100 datasets across various neural network models. The table presents the average and standard deviation of online accuracy achieved during a single epoch of online fine-tuning with different methods, where higher values indicate better performance. The boldfaced numbers represent the best values achieved for each column.

| Method | MNIST DenseNet-121 | CIFAR-10 ResNet18 | CIFAR-10 ViT-B16 | CIFAR-100 ResNet50 | CIFAR-100 ViT-B16 | ImageNet100 ViT-L16 |
|---|---|---|---|---|---|---|
| LoKO (ours) | $\mathbf{98.17_{\pm 0.56}}$ | $\mathbf{96.05_{\pm 0.12}}$ | $\mathbf{99.93_{\pm 0.05}}$ | $\mathbf{79.39_{\pm 0.12}}$ | $91.56_{\pm 0.40}$ | $97.71_{\pm 0.22}$ |
| LoRA/AdamW | $94.73_{\pm 0.65}$ | $65.98_{\pm 0.52}$ | $98.98_{\pm 0.49}$ | $60.39_{\pm 0.40}$ | $85.95_{\pm 0.40}$ | $95.13_{\pm 0.13}$ |
| LoRA/AdaGrad | $86.22_{\pm 4.10}$ | $61.83_{\pm 0.10}$ | $99.17_{\pm 0.20}$ | $40.62_{\pm 0.05}$ | $80.04_{\pm 0.87}$ | $93.91_{\pm 1.28}$ |
| DoRA/Kalman | — | — | $99.84_{\pm 0.03}$ | — | $\mathbf{92.17_{\pm 0.58}}$ | $\mathbf{97.96_{\pm 0.05}}$ |
| DoRA/AdamW | — | — | $98.82_{\pm 0.02}$ | — | $89.43_{\pm 0.18}$ | $94.50_{\pm 0.08}$ |
| DoRA/AdaGrad | — | — | $99.01_{\pm 0.04}$ | — | $85.39_{\pm 1.11}$ | $92.88_{\pm 0.06}$ |

Table 2: Results for SST-2, COLA, and MRPC datasets across various neural network models. The table presents the average and standard deviation of online accuracy achieved during online fine-tuning with different methods, where higher values indicate better performance. For SST-2, we conducted a single epoch of training. For COLA and MRPC, we performed multiple epochs due to their smaller dataset sizes. The bolded numbers represent the maximum values for each column.

| Method | SST-2 $RoB_{base}$ | SST-2 $RoB_{large}$ | COLA $RoB_{base}$ | COLA $RoB_{large}$ | MRPC $RoB_{base}$ | MRPC $RoB_{large}$ |
|---|---|---|---|---|---|---|
| LoKO (ours) | $88.41_{\pm 0.2}$ | $88.21_{\pm 0.1}$ | $83.67_{\pm 0.3}$ | $82.69_{\pm 0.4}$ | $89.17_{\pm 0.3}$ | $88.70_{\pm 0.6}$ |
| LoRA/AdamW | $89.55_{\pm 0.1}$ | $\mathbf{90.66_{\pm 0.2}}$ | $\mathbf{84.41_{\pm 0.1}}$ | $\mathbf{86.19_{\pm 0.9}}$ | $90.95_{\pm 0.3}$ | $93.04_{\pm 0.4}$ |
| LoRA/AdaGrad | $87.79_{\pm 0.2}$ | $86.25_{\pm 1.1}$ | $78.31_{\pm 0.0}$ | $79.72_{\pm 0.3}$ | $81.98_{\pm 0.1}$ | $83.88_{\pm 1.1}$ |
| DoRA/Kalman | $88.32_{\pm 0.2}$ | $88.76_{\pm 0.2}$ | $84.05_{\pm 0.0}$ | $83.33_{\pm 0.1}$ | $89.64_{\pm 0.34}$ | $89.49_{\pm 0.27}$ |
| DoRA/AdamW | $\mathbf{89.64_{\pm 0.1}}$ | $90.50_{\pm 0.6}$ | $84.37_{\pm 0.1}$ | $85.33_{\pm 1.5}$ | $\mathbf{91.48_{\pm 0.2}}$ | $\mathbf{93.57_{\pm 0.1}}$ |
| DoRA/AdaGrad | $87.99_{\pm 0.1}$ | $86.75_{\pm 0.7}$ | $78.29_{\pm 0.1}$ | $79.51_{\pm 0.7}$ | $82.02_{\pm 0.4}$ | $83.87_{\pm 0.0}$ |

dataset is employed for experimentation, utilizing the LeNet-5 architecture due to its fast training speed and low computational demands. Our experiments combine two proposed methods for initializing $\hat{p}_0$ with four widely used parameter initialization strategies: Xavier Uniform and Xavier Normal (Glorot & Bengio, 2010), as well as Kaiming Uniform and Kaiming Normal (He et al., 2015). Figure 6 in Appendix D illustrates the average online accuracy of the MNIST dataset after 1000 iterations of training from scratch. As shown, when $\hat{p}_0$ values are set far from the lower and upper bounds, the algorithm diverges, resulting in low prediction accuracy. The results indicate that initializing $\hat{p}_0$ with random values drawn from a uniform distribution (second method) provides a robust and wide range of choices for $\hat{p}_0$ without causing divergence. Our experiments demonstrate that as long as the initialization falls within an acceptable range, the filter will eventually converge to the optimal values. Notably, the optimal values are not highly sensitive to the initial covariance matrix. To obtain the upper bounds for our case studies, we conducted a series of experiments across a wide range of initial values and tracked the average online accuracy. If the accuracy doesn't reach a specific threshold within a predetermined number of iterations, we consider the test to have diverged. The boundary values derived from this analysis have been reported in Table 4 in Appendix D, specifically for case studies where an upper bound for the initial values of the covariance matrix is defined. To the best of our knowledge, no general criterion exists for this selection, and only a few studies have addressed the issue of initialization such as (Heimes, 1998; Rivals & Personnaz, 1998).

**Approximation of $R$:** Our proposed methods for approximating $R_k$ are presented in Table 5 in Appendix D alongside other estimation methods from the literature. Additionally, we have included the accuracy results on the MNIST test dataset for comparison.

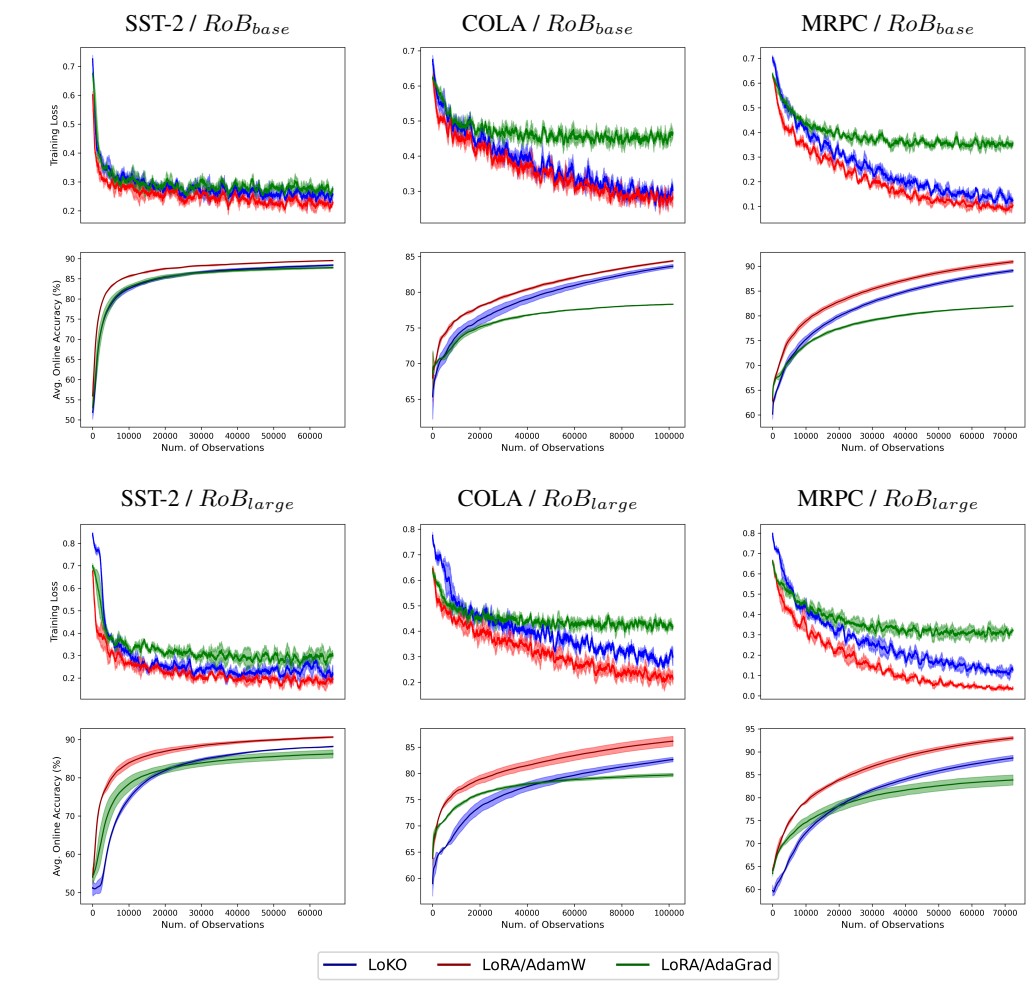

Figure 2: Comparison of LoKO (blue) with LoRA/AdamW (red) and LoRA/AdaGrad (green) across various language models and datasets. For each combination, the top row presents the training loss, while the bottom row illustrates the average online accuracy against the number of data points observed.

## 6    CONCLUSION

In this study, we present the Low-Rank Kalman Optimizer (LoKO), a Kalman-based training algorithm. LoKO offers a comparative alternative to advanced gradient-based optimizers for online fine-tuning scenarios. By leveraging the low-rank adaptation technique, and a diagonal approximation of the covariance matrix and incorporating a novel observation noise estimation technique based on EMA, we reformulate the EKF algorithm accordingly. Empirical evaluations on the benchmark of computer vision and language models, including MNIST, CIFAR-10/100, ImageNet100, SST-2, COLA, and MRPC demonstrate that LoKO consistently achieves superiority over (or at least comparability with) commonly used optimizers, showcasing its potential for efficient online fine-tuning of large models. Although this paper focuses on online fine-tuning in computer vision and language models for classification tasks, further evaluations are necessary across a broader range of domains, such as reinforcement learning. We leave these explorations for future work.

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

TECHNICAL APPENDICES

## A  LoKO ALGORITHM

The LoKO online fine-tuning method is outlined in Algorithm 1, with our modification to the EKF algorithm highlighted in red for clarity.

---

**Algorithm 1** LoKO

---

1:  **Initialization:**
2:  Define & initialize trainable parameters using LoRA: $\tilde{\boldsymbol{\theta}}_0$
3:  Initialize covariance: $\hat{\boldsymbol{p}}_0$
4:  Initialize matrix $\boldsymbol{R}$: $\boldsymbol{R}_0 = \boldsymbol{O}_{m \times m}$
5:
6:  **Online Fine-Tuning:**
7:  **while** new data available **do**
8:      **Get data:**
9:      data input: $\boldsymbol{x}_k$
10:     data output: $\boldsymbol{y}_k$
11:
12:     **Predict:**
13:     Predicted parameters: $\tilde{\boldsymbol{\theta}}_{k|k-1} = \tilde{\boldsymbol{\theta}}_{k-1}$
14:     Predicted covariance: $\hat{\boldsymbol{p}}_{k|k-1} = \hat{\boldsymbol{p}}_{k-1}$
15:
16:     **Pre-Updating:**
17:     Forward-propagation: $\hat{\boldsymbol{y}}_k = h_{LoRA}(\boldsymbol{x}_k, \tilde{\boldsymbol{\theta}}_{k|k-1})$
18:     Jacobian matrix: $\boldsymbol{H}_k = \frac{\partial h_{LoRA}}{\partial \tilde{\boldsymbol{\theta}}}\big|_{(\boldsymbol{x}_k, \tilde{\boldsymbol{\theta}}_{k|k-1})}$
19:     $\hat{\boldsymbol{R}}_k$ calculation: $\hat{\boldsymbol{R}}_k = (\boldsymbol{y}_k - \hat{\boldsymbol{y}}_k)(\boldsymbol{y}_k - \hat{\boldsymbol{y}}_k)^\top + \boldsymbol{H}_k(\hat{\boldsymbol{p}}_{k|k-1} \bullet \boldsymbol{H}_k^\top)$
20:     $\boldsymbol{R}_k$ estimation: $\boldsymbol{R}_k = \beta \boldsymbol{R}_{k-1} + (1-\beta)\hat{\boldsymbol{R}}_k$
21:
22:     **Update:**
23:     Compute Kalman gain: $\boldsymbol{K}_k = \hat{\boldsymbol{p}}_{k|k-1} \bullet \boldsymbol{H}_k^\top \left(\boldsymbol{H}_k(\hat{\boldsymbol{p}}_{k|k-1} \bullet \boldsymbol{H}_k^\top) + \boldsymbol{R}_k\right)^{-1}$
24:     Update parameters: $\tilde{\boldsymbol{\theta}}_k = \tilde{\boldsymbol{\theta}}_{k|k-1} + \boldsymbol{K}_k(\boldsymbol{y}_k - \hat{\boldsymbol{y}}_k)$
25:     Update covariance: $(\hat{\boldsymbol{p}}_k)^i = (\hat{\boldsymbol{p}}_{k|k-1})^i - (\boldsymbol{K}_k)^i_j (\boldsymbol{H}_k)^j_i (\hat{\boldsymbol{p}}_{k|k-1})^i$
26:
27:     **Output:**
28:     Updated Parameters: $\tilde{\boldsymbol{\theta}}_k$
29:  **end while**

---

## B  PROOFS AND DETAILED DERIVATIONS

This section presents the proofs and detailed derivations of the proposed LoKO algorithm, offering a comprehensive understanding of its underlying principles and mechanisms.

### B.1  PROOF FOR PROPOSITION 1

Here, we present the derivation of equation 11a and equation 11c from Proposition 1. These equations represent our proposed method for calculating the Kalman gain and updating the covariance matrix using a diagonal approximation.

**Proof of equation 11a** : Here, we will show that equation 11a is equivalent to equation 3a under the assumption of covariance diagonal approximation. For this aim, we need to show that $P_{k|k-1} H_k^\top = \hat{p}_{k|k-1} \bullet H_k^\top$.

*Proof.* For simplicity, let's drop the subscripts of $\hat{p}_{k|k-1}$, and $H_k^\top$. Now, define a diagonal matrix: $P = diag([p_1, p_2, ..., p_n])$, where the diagonal elements $[p_1, p_2, ..., p_n]$ are the elements of the vector $\hat{p}$, and its off-diagonal elements are zeros. Then, the $i^{th}$ row of $PH^\top$ can be represented as: $\left(PH^\top\right)_i = p_i \left(H^\top\right)_i$, where $\left(H^\top\right)_i$ represents the $i^{th}$ row of matrix $H^\top$. Now, let's represent the $i^{th}$ row of the transposed Khatri–Rao product of the vector $\hat{p}$ and matrix $H^\top$: $\left(\hat{p} \bullet H^\top\right)_i = (\hat{p})_i \otimes \left(H^\top\right)_i$. Since $(\hat{p})_i = p_i$, and $p_i$ is a scalar, the equality $p_i \otimes \left(H^\top\right)_i = p_i \left(H^\top\right)_i$ holds true. Consequently, $PH^\top = \hat{p} \bullet H^\top$. □

**Proof of equation 11c** : Here also, we will demonstrate that under the assumption of covariance diagonal approximation, the equation 11c is equivalent to equation 3c in vanilla EKF algorithm.

*Proof.* Again, by dropping the subscripts, let's define a diagonal matrix: $P$ whose diagonal elements are the elements of the vector $\hat{p}$. Then, let's expand the equation expressed with Einstein notation: $(K)_j^i (H)_i^j (\hat{p})^i = (K)_j^i (H)_i^j (P)_i^i = (KHP)_i^i$, where $(KHP)_i^i$ represents the $i^{th}$ element of the main diagonal of the matrix $KHP$. Consequently, the equation 11c represents the diagonal version of the covariance update in equation 3c. □

### B.2 PROOF FOR $R_k$ APPROXIMATION

Here, we derive Equation equation 7 and Equation equation 12, which corresponds to two different methods for estimating the matrix $R_k$.

**Proof of equation 7 for method 1:**

*Proof.* We start by the definition of the observation noise covariance matrix $v_k$: $R_k = \mathbb{E}[v_k v_k^T]$. Substituting the noise $v_k$ with $v_k = y_k - h_{LoRA}(x_k, \tilde{\theta}_k)$, we get: $R_k = \mathbb{E}\left\{\left(y_k - h_{LoRA}(\tilde{\theta}_k, x_k)\right)\left(y_k - h_{LoRA}(\tilde{\theta}_k, x_k)\right)^\top\right\}$. Assuming $\tilde{\theta}_k \approx \tilde{\theta}_{k|k-1}$, we will have:

$R_k = \mathbb{E}\left\{\left(y_k - h_{LoRA}(\tilde{\theta}_{k|k-1}, x_k)\right)\left(y_k - h_{LoRA}(\tilde{\theta}_{k|k-1}, x_k)\right)^\top\right\}$. To compute the expectation outlined above in an empirical manner, we define: $\hat{R}_k = \left(y_k - h_{LoRA}(\tilde{\theta}_{k|k-1}, x_k)\right)\left(y_k - h_{LoRA}(\tilde{\theta}_{k|k-1}, x_k)\right)^\top$ as the impact of new data on $R_k$, and employ an incremental averaging approach: $R_k = \frac{k-1}{k} R_{k-1} + \frac{1}{k}\hat{R}_k$. This incremental averaging can be expressed as an EMA approach: $R_k = \beta R_{k-1} + (1-\beta)\hat{R}_k$, with the forgetting factor of $\beta$. □

**Proof of equation 12 for method 2:**

*Proof.* Similar to method 1, we start with the definition of the covariance matrix of the observation noise $v_k$: $R_k = \mathbb{E}[v_k v_k^T]$. Based on $v_k = y_k - h_{LoRA}(x_k, \tilde{\theta}_k)$, and substituting the noise definition, we get: $R_k = \mathbb{E}\left\{\left(y_k - h_{LoRA}(\tilde{\theta}_k, x_k)\right)\left(y_k - h_{LoRA}(\tilde{\theta}_k, x_k)\right)^\top\right\}$. Since the value of $\tilde{\theta}_k$ is not available before *Updating* step, let's approximate $h_{LoRA}(\tilde{\theta}_k, x_k)$ using first-order Taylor series expansion around the last updated parameters $\tilde{\theta}_{k|k-1}$, which is: $h_{LoRA}(\tilde{\theta}_k, x_k) = h_{LoRA}(\tilde{\theta}_{k|k-1}, x_k) + H_k(\tilde{\theta}_k - \tilde{\theta}_{k|k-1})$. Thus: $R_k = $

$\mathbb{E}\left\{\left(y_k - h_{LoRA}(\tilde{\theta}_{k|k-1}, x_k) - H_k(\tilde{\theta}_k - \tilde{\theta}_{k|k-1})\right)\left(y_k - h_{LoRA}(\tilde{\theta}_{k|k-1}, x_k) - H_k(\tilde{\theta}_k - \tilde{\theta}_{k|k-1})\right)^\top\right\}$.

Expanding the above expression with dropping subscript and $(\tilde{\theta}_{k|k-1}, x_k)$ from

$h_{LoRA}(\tilde{\boldsymbol{\theta}}_{k|k-1}, \boldsymbol{x}_k)$, and using $\overline{\boldsymbol{\theta}}_k = (\tilde{\boldsymbol{\theta}}_k - \tilde{\boldsymbol{\theta}}_{k|k-1})$ for simplicity will yield: $\boldsymbol{R}_k =$
$\mathbb{E}\left\{ \boldsymbol{y}_k \boldsymbol{y}_k^\top - \boldsymbol{y}_k h^\top - \boldsymbol{y}_k \overline{\boldsymbol{\theta}}_k^\top \boldsymbol{H}_k^T - h\boldsymbol{y}_k^\top + hh^\top + h\overline{\boldsymbol{\theta}}_k^\top \boldsymbol{H}_k^T - \boldsymbol{H}_k \overline{\boldsymbol{\theta}}_k \boldsymbol{y}_k^\top + \boldsymbol{H}_k \overline{\boldsymbol{\theta}}_k h^\top + \boldsymbol{H}_k \overline{\boldsymbol{\theta}}_k \overline{\boldsymbol{\theta}}_k^\top \boldsymbol{H}_k^T \right\}$.
Now, let's take the expectation by considering:
$\mathbb{E}[\overline{\boldsymbol{\theta}}_k] = \mathbb{E}[\tilde{\boldsymbol{\theta}}_k - \tilde{\boldsymbol{\theta}}_{k|k-1}] = 0$, and $\mathbb{E}[\overline{\boldsymbol{\theta}}_k \overline{\boldsymbol{\theta}}_k^\top] = \mathbb{E}[(\tilde{\boldsymbol{\theta}}_k - \tilde{\boldsymbol{\theta}}_{k|k-1})(\tilde{\boldsymbol{\theta}}_k - \tilde{\boldsymbol{\theta}}_{k|k-1})^\top] = \boldsymbol{P}_{k|k-1}$.
Now, we have: $\boldsymbol{R}_k = \mathbb{E}\left\{ \boldsymbol{y}_k \boldsymbol{y}_k^\top - \boldsymbol{y}_k h^\top - h\boldsymbol{y}_k^\top + hh^\top + \boldsymbol{H}_k \boldsymbol{P}_{k|k-1} \boldsymbol{H}_k^T \right\}$. We
already knew that $\boldsymbol{H}_k \boldsymbol{P}_{k|k-1} \boldsymbol{H}_k^T = \boldsymbol{H}_k(\hat{\boldsymbol{p}}_{k|k-1} \bullet \boldsymbol{H}_k^\top)$. Thus: $\boldsymbol{R}_k =$
$\mathbb{E}\left\{ (\boldsymbol{y}_k - h)(\boldsymbol{y}_k - h)^\top + \boldsymbol{H}_k(\hat{\boldsymbol{p}}_{k|k-1} \bullet \boldsymbol{H}_k^\top) \right\}$. Similar to the previous approach,
we compute the above expectation in an empirical manner by defining: $\hat{\boldsymbol{R}}_k =$
$\left( \boldsymbol{y}_k - h_{LoRA}(\tilde{\boldsymbol{\theta}}_{k|k-1}, \boldsymbol{x}_k) \right) \left( \boldsymbol{y}_k - h_{LoRA}(\tilde{\boldsymbol{\theta}}_{k|k-1}, \boldsymbol{x}_k) \right)^\top + \boldsymbol{H}_k(\hat{\boldsymbol{p}}_{k|k-1} \bullet \boldsymbol{H}_k^\top)$

$\square$

## C  COMPUTATIONAL ANALYSIS

### C.1  COMPUTATIONAL COMPLEXITY

To analyze the computational complexity of our proposed algorithm, let $n$ represent the number of parameters, $\tilde{n}$ the number of trainable parameters ($\tilde{n} \ll n$), and $m$ the number of model outputs. The computational complexities of each step in the LoKO algorithm compared to the vanilla Kalman filter are as follows:

**Prediction:**  The computational complexity of both LoKO and the vanilla Kalman filter for the Prediction step is $\mathcal{O}(1)$.

**Pre-Updating:**  For LoKO, estimating the observation noise covariance using equation 7 has a complexity of $\mathcal{O}(m^2)$. When using equation 12, the complexity increases to $\mathcal{O}(m^2 \tilde{n})$. In contrast, for the vanilla Kalman filter, the complexity for equation 7 is $\mathcal{O}(m^2)$, while for equation 12 it is $\mathcal{O}(m^2 n + n^2 m)$.

**Updating:**  The computational complexity for LoKO when calculating the Kalman gain ( equation 11a) is $\mathcal{O}(m^3 + m^2 \tilde{n})$, while for the vanilla Kalman filter (using equation 3a), it will be $\mathcal{O}(m^3 + m^2 n + n^2 m)$. For updating parameters (equation 11b), LoKO has a complexity of $\mathcal{O}(m\tilde{n})$, compared to $\mathcal{O}(mn)$ for the vanilla Kalman filter. And finally, the complexity of covariance updating (equation 11c) in LoKO is $\mathcal{O}(\tilde{n})$, while for the vanilla Kalman filter (using equation 3c), it will be $\mathcal{O}(n^2 m)$.
Thus, the total computational complexity in the worst-case scenario, for LoKO is $\mathcal{O}(m^3 + m^2 \tilde{n})$, and for the vanilla Kalman filter will be $\mathcal{O}(m^3 + m^2 n + n^2 m)$. In cases where the number of parameters is significantly larger than the output size, the dominant term for LoKO is $\mathcal{O}(m^2 \tilde{n})$, while for the vanilla Kalman filter, it will be $\mathcal{O}(n^2 m)$. Therefore, LoKO reduces the computational complexity from quadratic to linear in the number of parameters as well as decreasing the number of trainable parameters ($\tilde{n} \ll n$).

### C.2  TIME ANALYSIS

To evaluate the time efficiency of our LoKO algorithm, we compare the number of steps required for convergence, similar to the criterion used in (Liu et al., 2023). Convergence is defined as the number of iterations at which the loss stays flat or decreases by less than a specified threshold. The time analysis has been presented in Table 3

Table 3: Time efficiency of LoKO in comparison to the baseline methods.

| Dataset - Model | LoKO | | LoRA/AdamW | | LoRA/AdaGrad | |
|---|---|---|---|---|---|---|
| | steps | per-step time | steps | per-step time | steps | per-step time |
| MNIST - DenseNet-121 | 4000 | 0.23s | 20000 | 0.025s | 23000 | 0.026s |
| CIFAR10 - ResNet18 | 20000 | 0.11s | 30000 | 0.007s | 35000 | 0.006s |
| CIFAR10 - ViT-B16 | 2000 | 0.14s | 2500 | 0.013s | 4000 | 0.015s |
| CIFAR100 - ResNet50 | 30000 | 0.67s | 30000 | 0.017s | 20000 | 0.018s |
| CIFAR100 - ViT-B16 | 2000 | 0.53s | 7000 | 0.014s | 15000 | 0.015s |
| ImageNet100 - ViT-L16 | 5000 | 1.4s | 12000 | 0.03s | 25000 | 0.032s |
| SST-2 - RoBbase | 48000 | 0.139s | 47000 | 0.022s | 48000 | 0.02s |
| SST-2 - RoBlarge | 40000 | 0.17s | 45000 | 0.034s | 42000 | 0.034 |
| COLA - RoBbase | 87000 | 0.137s | 87000 | 0.02s | 45000 | 0.022s |
| COLA - RoBlarge | 100000 | 0.169s | 100000 | 0.035s | 50000 | 0.034s |
| MRPC - RoBbase | 60000 | 0.139s | 60000 | 0.02s | 35000 | 0.021s |
| MRPC - RoBlarge | 65000 | 0.169s | 60000 | 0.036s | 46000 | 0.035s |

# D  ADDITIONAL EXPERIMENT DETAILS

## D.1  RESULTS FOR DoRA EXPERIMENTS

Figure 3 and 4 show the results for DoRA experiments on vision and language datasets, respectively.

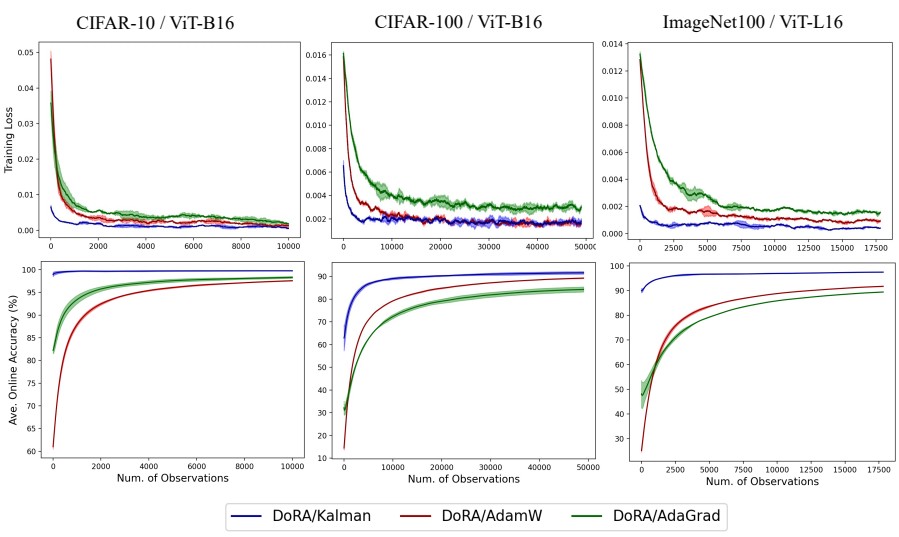

Figure 3: Performance of DoRA/Kalman (blue) compared to DoRA/AdamW (red) and DoRA/AdaGrad (green) for different computer vision datasets and models. The upper rows show the training loss, and the lower rows display the average online accuracy versus the number of observed data.

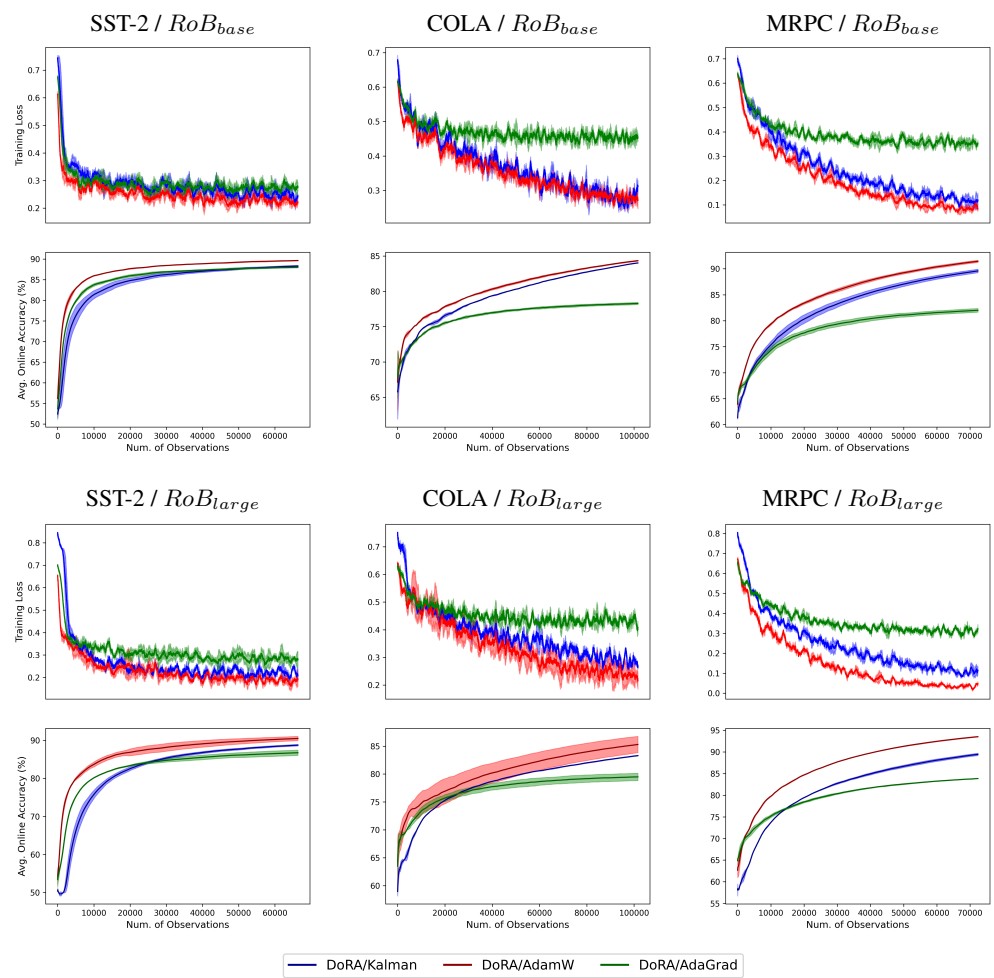

Figure 4: Comparison of DoRA/Kalman (blue) with DoRA/AdamW (red) and DoRA/AdaGrad (green) across various language models and datasets. For each combination, the top row presents the training loss, while the bottom row illustrates the average online accuracy against the number of data points observed.

## D.2 DIAGONAL APPROXIMATION OF COVARIANCE MATRIX

Our empirical observations reveal that throughout the training process, the covariance matrix of a feedforward neural network tends towards a (block-)diagonal structure asymptotically. Figure 5 illustrates the evolution of the covariance matrix $P_k \in \mathbb{R}^{n \times n}$ in LeNet-5 utilizing the Kalman optimizer. Initially, the matrix exhibits a fully dense positive-definite form, which progressively transitions towards a (block-)diagonal configuration as the training algorithm advances.

## D.3 INITIALIZATION OF $\hat{p}_0$

**Ablation of $\hat{p}_0$ Initialization:** The initialization of $\hat{p}_0$ with our two methods for MNIST/LeNet-5 has been showed in Figure 6.

The lower and upper bounds for the two proposed initialization methods of $\hat{p}_0$ have been reported in Table 4 for case studies where an upper bound for the initial values of the covariance matrix is defined.

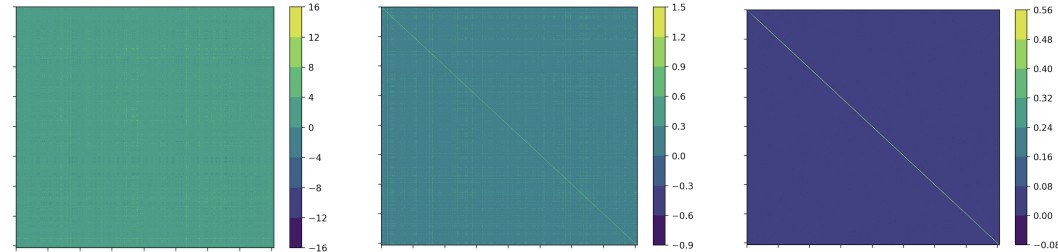

Figure 5: Evolution of covariance matrix $P_k \in \mathbb{R}^{n \times n}$ in LeNet-5 using Kalman optimizer. The matrix starts with a fully dense positive-definite matrix, and with the progress of the training algorithm, it gradually converges to a (block-)diagonal configuration.

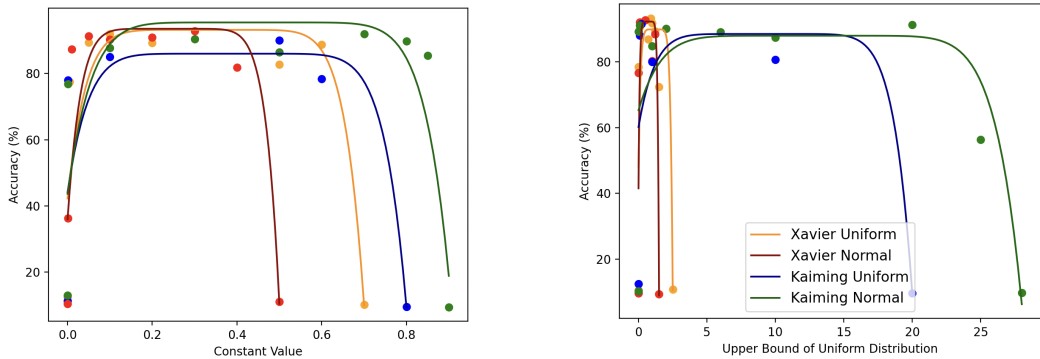

Figure 6: Initialization of $\hat{p}_0$ with two different techniques: (**left**) Setting a constant positive value, and (**right**) Assigning random positive values drawn from a uniform distribution.

Table 4: The lower and upper bounds for two proposed initialization methods of $\hat{p}_0$

| Dataset - Model | method 1 | | method 2 | |
|---|---|---|---|---|
| | min | max | min | max |
| MNIST - DenseNet-121 | 0.0001 | $0.11_{\pm 0.01}$ | 0.0001 | $0.32_{\pm 0.01}$ |
| CIFAR10 - ViT-B16 | 0.001 | $105_{\pm 3}$ | 0.001 | $165_{\pm 3}$ |
| CIFAR100 - ViT-B16 | 0.001 | $0.2_{\pm 0.01}$ | 0.001 | $0.59_{\pm 0.01}$ |
| ImageNet100 - ViT-L16 | 0.001 | $1.0_{\pm 0.1}$ | 0.001 | $2.0_{\pm 0.1}$ |

**Sensitivity to Initialization of $\hat{p}_0$:** As explained in Section 5.3, the optimal values achieved by LoKO are not highly sensitive to the initial covariance matrix. To show this, we present evidence based on the results of CIFAR-100/ViT-B16 for four different covariance initialization values. As demonstrated in Figure 7, the final outcomes show minimal sensitivity to these initializations.

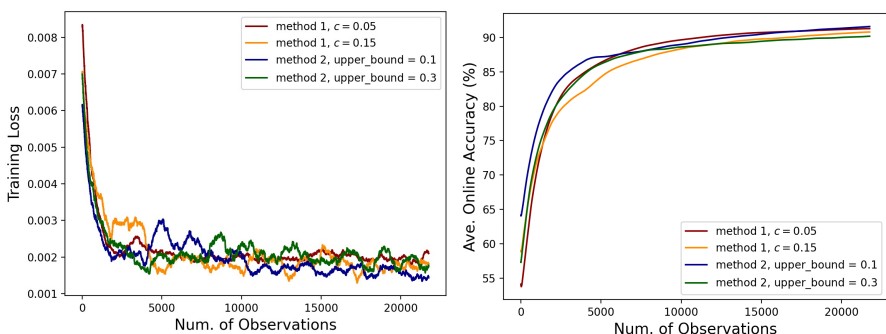

Figure 7: Sensitivity analysis to the initial values of $\hat{\boldsymbol{p}}_0$

## D.4 ESTIMATION OF $R$

A comparison of different approaches in the estimation of the observation noise covariance matrix has been provided in Table 5.

Table 5: Different approaches in the approximation of the observation noise covariance matrix

| Ref. | Equation | Accuracy MNIST |
|---|---|---|
| Puskorius & Feldkamp (1991) Murtuza & Chorian (1994) | $R_k = I$ | 92.59 % |
| Singhal & Wu (1988) Singhal & Wu (1989) | $R_k = I e^{-k/50}$ | 93.65 % |
| Ollivier (2018) Chang et al. (2022) | $R_k = diag(\hat{\mathbf{y}}_k) - \hat{\mathbf{y}}_k \hat{\mathbf{y}}_k^\top$ | 93.89 % |
| LoKO (method 1) | $R_k = \beta R_{k-1} + (1-\beta)\hat{R}_k,$ $\hat{R}_k = (\mathbf{y}_k - \hat{\mathbf{y}}_k)(\mathbf{y}_k - \hat{\mathbf{y}}_k)^\top$ | 93.81 % |
| LoKO (method 2) | $R_k = \beta R_{k-1} + (1-\beta)\hat{R}_k,$ $\hat{R}_k = (\mathbf{y}_k - \hat{\mathbf{y}}_k)(\mathbf{y}_k - \hat{\mathbf{y}}_k)^\top + H_k(\hat{p}_{k|k-1} \bullet H_k^\top)$ | **94.51** % |

## D.5 SENSITIVITY TO $\beta$ IN EMA

To assess the impact of hyperparameter $\beta$ on LoKO's performance, we conducted a sensitivity analysis varying $\beta$ values. Through this analysis, we measured the average online accuracy across different $\beta$ settings. The results illustrate that excessively high or low $\beta$ values notably compromise LoKO's performance. Optimal performance lies within the range of 0.9 to 0.98 for $\beta$. See Figure 8 for more details.

## D.6 SENSITIVITY TO OOD DATA

To evaluate the sensitivity of LoKO to the Out-of-Distribution (OOD) data, we conducted a set of experiments on MNIST classification task in an online manner. To generate the out-of-distributed MNIST images, we applied a combination of random rotation (30 degrees) and color jitter (with a brightness and contrast adjustment of 0.5). Then, we inserted these OOD images into the online fine-tuning process by including one OOD sample after every 100 normal samples. As shown in Figure 9, both LoKO and AdamW exhibit some sensitivity to OOD data due to their use of the EMA technique. However, the results for LoKO demonstrate that it adapts more quickly to the shifted distribution compared to AdamW. In contrast, AdaGrad shows less sensitivity to OOD data, as it does not incorporate the EMA technique for the first moment.

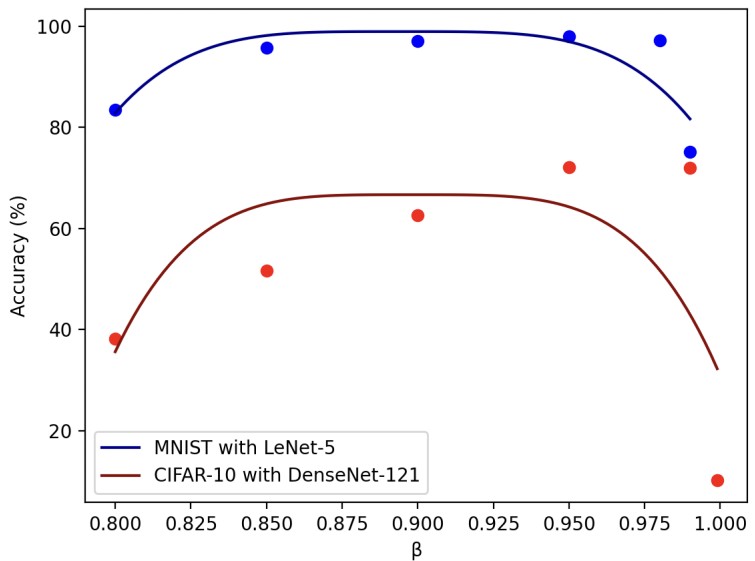

Figure 8: Sensitivity to $\beta$

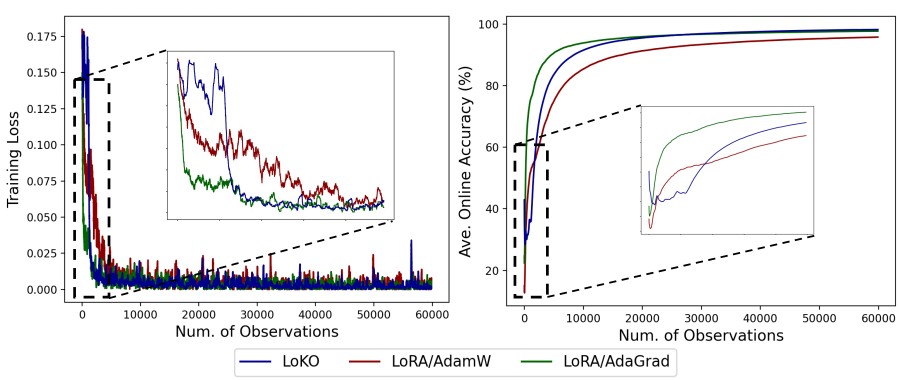

Figure 9: Sensitivity analysis to the OOD data

# E ADDITIONAL INFORMATION

The Table 6 shows the LoRA layers, number of total parameters, and number of trainable parameters in our experiments.

Table 6: Total and trainable parameters for each model utilized in the experiments.

| Model | LoRA layer | Num. of total parameters | Num. of trainable parameters |
|---|---|---|---|
| DenseNet-121 | convolution | 7.5 M | 166 K |
| ResNet18 | convolution | 11 M | 150 K |
| ViT-B16 | query | 86 M | 155 K |
| ResNet50 | convolution | 24 M | 520 K |
| ViT-L16 | query | 305 M | 400 K |
| RoBERTa-base | query | 125 M | 666 K |
| RoBERTa-large | query | 355 M | 1248 K |

