# OpenReview forum: "LoKO: Low-Rank Kalman Optimizer for Online Fine-Tuning of Large Models"
_ICLR.cc/2025/Conference — Submitted to ICLR 2025_

### Official Review · Reviewer_Y8Hg · 2024-10-26

**Soundness:** 3
**Presentation:** 3
**Contribution:** 2
**Rating:** 5
**Confidence:** 3

**Summary:**

This paper proposes an innovative Kalman filter-based low-rank optimization algorithm (referred to as LoKO) aimed at online fine-tuning of large-scale models.

The algorithm initially employs the LoRA algorithm to effectively reduce the number of trainable parameters through low-rank decomposition.
Subsequently, it adopts a diagonal approximation of the covariance matrix P and integrates Exponential Moving Average (EMA) techniques to estimate the covariance matrix R of the observation noise.

The implementation of these strategies significantly lowers the computational complexity of LoKO. Moreover, experimental results across
multiple computer vision and language models demonstrate that its performance is on par with existing advanced methods.

**Strengths:**

This paper innovatively proposes using the Kalman Filter algorithm combined with LoRA for online fine-tuning of large-scale models, and
reduces the computational complexity of fine-tuning through approximate solution methods.

The experiments also demonstrate that the proposed method achieves performance comparable to existing gradient descent-based algorithms across multiple models and datasets.

Finally, the paper investigates the sensitivity of the final performance to the initialization range of hyperparameters through experiments, thereby proving the robustness of the proposed method.

**Weaknesses:**

The LoKO method assumes that the model's trainable parameters remain unchanged during the observation process, meaning that the generation process of the observed variables remains consistent throughout the entire observation period. This assumption is often difficult to satisfy in online scenarios and results in the loss of the model's dynamic adaptation capability.

Although the authors observed through experiments that the matrix tends to become diagonal during training, they did not provide a
corresponding analysis.

In Table 3, the LoKO method consistently underperforms compared to the baseline methods LoRA/AdamW, but the authors did not provide any related analysis.

One significant contribution of this paper is the proposal of a method for online fine-tuning of large-scale models. However, the largest model tested in the experiments, RoBERTa-large, has only 335M parameters. Therefore, further testing on even larger models is needed to validate the effectiveness of their method.

Finally, the paper lacks a comparison of the runtime efficiency between LoKO and the baseline methods.

**Questions:**

In Table 1, the authors provide the tested models and their corresponding LoRA layers along with the trainable parameters. How do the remaining non-trainable parameters be obtained?

In Table 4, the authors provide the diagonal initialization bounds for parameter P for different datasets and models. How are these bounds determined? Is there a more general selection criterion?

---

> ### Author Response · Authors · 2024-11-22
> **General Comment**
>
> First of all, we sincerely appreciate your invaluable feedback. We have made every effort to address your concerns thoroughly, and below, we provide a detailed point-by-point response to each of your comments. Furthermore, all modifications have been clearly highlighted in the revised version of the paper that has been submitted.

---

> ### Author Response · Authors · 2024-11-22
> **Dynamic Adaptation**
>
> We appreciate your comment and acknowledge the importance of studying dynamic adaptation. However, our work primarily focuses on the development and evaluation of the LoKO algorithm within the stationary dataset. As discussed in [1],  dynamic adaptation typically is valid in the context of Online Continual Learning (OCL) with non-stationary data, which lies outside the primary focus of our current work. We believe that the current methodology is appropriate for the setting and domain considered in our experiments. Nonetheless, we recognize the importance of this issue and consider it an interesting avenue for future work.

---

> ### Author Response · Authors · 2024-11-22
> **Diagonal Approximation**
>
> As mentioned in response to other reviewers, the adoption of a diagonal approximation for large-scale covariance and Hessian matrices is a widely accepted practice in machine learning due to its computational efficiency. For instance,  [2] utilizes a diagonal Hessian approximation in the development of a second-order optimizer. Or, [3] demonstrates that the Adam algorithm effectively employs a diagonal approximation of the pure Newton’s method update. In addition, although the idea is interesting, the primary focus of this study is on the empirical investigation of LoKO's performance, rather than a detailed theoretical analysis of this approximation. However, we acknowledge the value of such analysis and consider it a promising direction for future work. One possible approach can be analyzing the evolution of the covariance matrix during Kalman iterations. By propagating the covariance matrix through the update equations, we should be able to somehow show that the off-diagonal elements converge to zero.

---

> ### Author Response · Authors · 2024-11-22
> **Language Tasks Results**
>
> Thank you for this insightful comment. As discussed in [4], AdamW optimizer benefits from the decoupled weight decay, which has contributed to its strong performance in language modeling tasks. On the other hand, the gradient-free Kalman algorithm used in LoKO does not incorporate a similar regularization mechanism, which may explain the slightly lower performance in certain cases compared to AdamW. We will clarify this point by adding the following discussion to Section 5.2 of the revised version of the paper:
>
> “As can be seen in Table 2, LoKO demonstrates slightly lower performance than AdamW on certain language tasks. A justification for this observation is the fact that AdamW optimizer benefits from the decoupled weight decay, which has contributed to its strong performance in language modeling tasks [4]. On the other hand, the gradient-free Kalman algorithm used in LoKO does not incorporate a similar regularization mechanism, which may explain the slightly lower performance in certain cases compared to AdamW.”

---

> ### Author Response · Authors · 2024-11-22
> **Large-scale Model**
>
> As detailed in Section 5, our experiments cover a broad range of computer vision and language models applied to various datasets for classification tasks. The models used in our study are well-known and widely recognized within the classification domain. We believe that the current adopted models are appropriate for the task considered in our experiments. However, to address your concern more effectively, we would kindly ask if you could specify particular models that you believe should be investigated within the context of this paper.

---

> ### Author Response · Authors · 2024-11-22
> **Computational Analysis**
>
> Thank you for your constructive feedback. As discussed in Section 4.1, applying a diagonal approximation for the covariance matrix, combined with a low-rank decomposition approach, decrease the computational complexity from quadratic to linear in the number of trainable parameters. To clarify this further, we will include the following explanation in Appendix C.1 of the revised paper:
>
> “Computational Analysis:
> Let n represent the number of parameters, n ̃ the number of trainable parameters (n ̃  ≪n), and m the number of model outputs. The computational complexities of each step in the LoKO algorithm compared to the vanilla Kalman filter are as follows:
> Prediction: The computational complexity of both LoKO and the vanilla Kalman filter for the Prediction step is O(1).
> Pre-Updating: For LoKO, estimating the observation noise covariance using Equation (7) has a complexity of O(m^2). When using Equation (12), the complexity increases to O(m^2 n ̃). In contrast, for the vanilla Kalman filter, the complexity for Equation (7) is O(m^2), while for Equation (12) it is O(m^2 n+n^2 m).
> Updating: The computational complexity for LoKO when calculating the Kalman gain (Equation (11a)) is O(m^3+m^2 n ̃), while for the vanilla Kalman filter (using Equation (3a)), it will be O(m^3+m^2 n+n^2 m). For updating parameters (Equation (11b)), LoKO has a complexity of O(mn ̃), compared to O(mn) for the vanilla Kalman filter. And finally, the complexity for covariance updating (Equation (11c)) in LoKO is O(n ̃), while for the vanilla Kalman filter (using Equation (3c)), it will be O(n^2 m).
> Thus, the total computational complexity in the worst-case scenario, for LoKO is O(m^3+m^2 n ̃), and for the vanilla Kalman filter will be O(m^3+m^2 n+n^2 m). In cases where the number of parameters is significantly larger than output size, the dominant term for LoKO is O(m^2 n ̃), while for the vanilla Kalman filter, it will be O(n^2 m). Therefore, LoKO reduces the computational complexity from quadratic to linear in the number of parameters as well as decreasing the number of trainable parameters (n ̃  ≪n).”
>
> To evaluate the time efficiency of our LoKO algorithm, we compare the number of steps required for convergence, similar to the criterion used in [2]. Convergence is defined as the number of iterations at which the loss stays flat or decreases by less than a specified threshold. The time analysis will be presented as follows in the Appendix C.2 of the revised paper.

---

> ### Author Response · Authors · 2024-11-22
> **Non-trainable Parameters**
>
> As discussed in [5] and explained in the manuscript, the widespread use of deep neural networks, particularly large models, is largely driven by pre-training on extensive datasets followed by task-specific fine-tuning. The non-trainable parameters have been obtained from different pretrained models. For MNIST/DenseNet-121, we used a backbone pretrained on ImageNet via PyTorch Image Models (timm) [6]. For CIFAR-10/ResNet18 and CIFAR-100/ResNet50, we employed pretrained models from [7], which were trained on ImageNet as the backbone. For CIFAR-10/ViT-B16, CIFAR-100/ViT-B16, and ImageNet100/ViT-L16, we used DINOv2 [8], which was pretrained on ImageNet. And, for the language tasks we employed RoBERTa pretrained model from [9]. We will clarify your point by adding the following discussion to the revised version of the paper:
>
> “We employ the following pre-trained models as backbones for our fine-tuning experiments:
>
> o	For MNIST/DenseNet-121, we used a backbone pretrained on ImageNet via PyTorch Image Models (timm) [6].
>
> o	For CIFAR-10/ResNet18 and CIFAR-100/ResNet50, we employed pretrained models from [7], which were trained on ImageNet as the backbone.
>
> o	For CIFAR-10/ViT-B16, CIFAR-100/ViT-B16, and ImageNet100/ViT-L16, we used DINOv2 [8], which was pretrained on ImageNet.
>
> o	For the language tasks we employed RoBERTa pretrained model from [9].”

---

> ### Author Response · Authors · 2024-11-22
> **Covariance Initialization**
>
> As mentioned in response to other reviewers, initializing the covariance matrix requires prior knowledge of the network weights and their associated uncertainties. In the absence of such knowledge, almost any initial values for the covariance matrix can be chosen, provided they are not too close to zero, and in some cases, they are not too far from an upper bound. As long as the initialization falls within an acceptable range, the filter will eventually converge to the optimal values. Notably, the optimal values are not sensitive to the initial covariance matrix. As discussed in detail in Section 5.3 of the paper; to obtain the upper bounds for our case studies, we conducted a series of experiments across a wide range of initial values and tracked the average online accuracy. If the accuracy doesn’t reach to a specific threshold within a predetermined number of iterations, we consider the test to have diverged. The boundary values derived from this analysis have been reported in Table 4 of the paper, specifically for case studies where an upper bound for the initial values of covariance matrix is defined. To the best of our knowledge, no general criterion exists for this selection, and only a few studies have addressed the issue of initialization such as [10] and [11]. To clarify more, we will add the following discussion to the revised version of the paper:
>
> “Initializing p ̂ requires prior knowledge of the network weights and their associated uncertainties. In the absence of such knowledge, almost any initial values for p ̂ can be chosen, provided they are not too close to zero, and in some cases, they are not too far from an upper bound. Our experiments demonstrate that as long as the initialization falls within an acceptable range, the filter will eventually converge to the optimal values. Notably, the optimal values are not sensitive to the initial covariance matrix. To obtain the upper bounds for our case studies, we conducted a series of experiments across a wide range of initial values and tracked the average online accuracy. If the accuracy doesn’t reach to a specific threshold within a predetermined number of iterations, we consider the test to have diverged. The boundary values derived from this analysis have been reported in Table 4, specifically for case studies where an upper bound for the initial values of covariance matrix is defined. To the best of our knowledge, no general criterion exists for this selection, and only a few studies have addressed the issue of initialization such as [10] and [11].”

---

> ### Author Response · Authors · 2024-11-22
> **References**
>
> [1]	M. K. Titsias, A. Galashov, A. Rannen-Triki, R. Pascanu, Y. W. Teh, and J. Bornschein, “KALMAN FILTER FOR ONLINE CLASSIFICATION OF NON-STATIONARY DATA,” 2024.
>
> [2]	H. Liu, Z. Li, D. Hall, P. Liang, and T. Ma, “Sophia: A Scalable Stochastic Second-order Optimizer for Language Model Pre-training,” Mar. 05, 2024, arXiv: arXiv:2305.14342. Accessed: Mar. 12, 2024. [Online]. Available: http://arxiv.org/abs/2305.14342
>
> [3]	I. Molybog et al., “A Theory on Adam Instability in Large-Scale Machine Learning,” Apr. 25, 2023, arXiv: arXiv:2304.09871. Accessed: Nov. 16, 2024. [Online]. Available: http://arxiv.org/abs/2304.09871
>
> [4]	S. Xie and Z. Li, “Implicit Bias of AdamW: $\ell_∞$-Norm Constrained Optimization,” in Proceedings of the 41st International Conference on Machine Learning, PMLR, Jul. 2024, pp. 54488–54510. Accessed: Nov. 17, 2024. [Online]. Available: https://proceedings.mlr.press/v235/xie24e.html
>
> [5]	Z. Han, C. Gao, J. Liu, J. Zhang, and S. Q. Zhang, “Parameter-Efficient Fine-Tuning for Large Models: A Comprehensive Survey,” Jul. 12, 2024, arXiv: arXiv:2403.14608. Accessed: Sep. 08, 2024. [Online]. Available: http://arxiv.org/abs/2403.14608
>
> [6]	R. Wightman et al., rwightman/pytorch-image-models: v0.8.10dev0 Release. (Feb. 07, 2023). Zenodo. doi: 10.5281/ZENODO.4414861.
>
> [7]	K. He, X. Zhang, S. Ren, and J. Sun, “Deep Residual Learning for Image Recognition,” Dec. 10, 2015, arXiv: arXiv:1512.03385. Accessed: Nov. 18, 2024. [Online]. Available: http://arxiv.org/abs/1512.03385
>
> [8]	M. Oquab et al., “DINOv2: Learning Robust Visual Features without Supervision,” Feb. 02, 2024, arXiv: arXiv:2304.07193. Accessed: Nov. 18, 2024. [Online]. Available: http://arxiv.org/abs/2304.07193
>
> [9]	Y. Liu et al., “RoBERTa: A Robustly Optimized BERT Pretraining Approach,” Jul. 26, 2019, arXiv: arXiv:1907.11692. Accessed: Nov. 18, 2024. [Online]. Available: http://arxiv.org/abs/1907.11692
>
> [10]	F. Heimes, “Extended Kalman filter neural network training: experimental results and algorithm improvements,” in SMC’98 Conference Proceedings. 1998 IEEE International Conference on Systems, Man, and Cybernetics (Cat. No.98CH36218), Oct. 1998, pp. 1639–1644 vol.2. doi: 10.1109/ICSMC.1998.728124.
>
> [11]	I. Rivals and L. Personnaz, “A recursive algorithm based on the extended Kalman filter for the training of feedforward neural models,” Neurocomputing, vol. 20, no. 1, pp. 279–294, Aug. 1998, doi: 10.1016/S0925-2312(98)00021-6.

---

### Official Review · Reviewer_WNJR · 2024-10-30

**Soundness:** 4
**Presentation:** 3
**Contribution:** 3
**Rating:** 6
**Confidence:** 4

**Summary:**

Previous works show that the recursive Extended Kalman Filter (EKF) algorithm can optimize relatively small models with performance surpassing its gradient-based counterparts. However, it is challenging to optimize neural networks with EKF because the size of the crucial covariance matrix in EKF can grow quadratically with the number of model parameters in training. In this paper, the core idea is to have EKF work with the LORA so that only a few parameters are involved in the EKF optimization. The proposed approach LoKO shows promising results on computer vision and language modeling benchmarks, such as MNIST, CIFAR 10/100, ImageNet 100, SST-2, COLA, MRPC.

**Strengths:**

LoKO seems to be an interesting approach that successfully combine the EKF and the LORA, this provides a good new perspective to approach the online gradient-descent method.

This paper provides a well structured experiments across computer vision and NLP. The results in general are not state-of-the-art, but they are decent to show the promising results of the LoKO.

The proof to show that the proposed approach equation is equivalent to that in the vanilla EKF algorithm seems to be sound.

The paper is well structured and well written. The logical flow from problem definition through methodology to empirical validation helps readers grasp the paper’s objectives and contributions effectively.

The proposed LoKO seems to provide high accuracy with lower computational footprint. This seems to be interesting to some practical applications where the computational efficiency is important.

**Weaknesses:**

The evaluation is limited. This paper has done quite a few evaluation on the proposed LoKO method. However, i find none of those datasets are well suited for the LoKO. In general, the model full fine-tuning can do better than that from the LoRA fine-tuning. The LoKO would be suboptimal compared to the model full-fining as long as one can afford the GPU computes. The proposed experiments do not cover the dynamic streaming type of data, which would be able to show the strength of the proposed LoKO.

The proposed LoKO seems to suggest to require one to initialize the covariance matrix, and noise estimation parameters. However, it is unclear how sensitive of those parameters to the final model performance. There are no studies to reveal the sensitive of the proposed approach.

There are multiple LoRA and its variants for PEFT. The experiments do not cover the comparison between the proposed one against the other PEFT methods such as QLoRA, QA-LoRA, LoftQ. It would be interesting and useful to see how the proposed LoKO compared against the other LoRA and its variants.

The experiments do not cover the study on the memory efficiency and what's the limitations of the proposed LoKO. It is important to understand the limits and boundary of the LoKO.

**Questions:**

1. What's the limitation of the proposed approach? It would be great to have more throughly memory usage and efficiency analysis and experiments, given the computational efficiency is one of the advantages of the proposed approach.

2. How to initialize the LoKO? how sensitive of the model is? It would be great to have more studies on those.

3. How does the proposed LoKO compare against the standard LoRA and other LoRA variants such as QLoRA?

---

> ### Author Response · Authors · 2024-11-22
> **General Comment**
>
> First of all, we sincerely thank you for the invaluable feedback. Our utmost effort has been exerted in addressing your concerns, and we have subsequently provided a comprehensive point-by-point explanation in the following list of responses. Furthermore, all modifications have been clearly highlighted in the revised version of the paper that has been submitted.

---

> ### Author Response · Authors · 2024-11-22
> **Evaluation and other LoRA variants**
>
> Thank you for this valuable suggestion. We find your recommendation highly insightful. We have conducted additional experiments involving the DoRA [1] variant of LoRA, in combination with Kalman, AdamW, and AdaGrad. The results from these experiments are now included in the revised paper, with visualizations in Appendix D.1 (Figure 3, and Figure 4) and further analysis in Table 1, and Table 2, Section 5.2. As shown, the results are promising for the Kalman algorithm, demonstrating its compatibility and effectiveness when integrated with this Weight-Decomposed Low-Rank Adaptation technique.

---

> ### Author Response · Authors · 2024-11-22
> **Full Fine-Tuning**
>
> As correctly pointed out in your comment, the primary limitation of full fine-tuning is the high computational cost, which can be prohibitive in many scenarios. Even with high GPU resources, based on discussion in [2], full fine-tuning can sometimes lead to catastrophic forgetting, which negatively impacts the model's ability to generalize effectively. LoKO, in contrast, aims to provide a more computationally efficient alternative while mitigating such issues.

---

> ### Author Response · Authors · 2024-11-22
> **Dynamic Streaming Type of Data**
>
> We appreciate your comment and acknowledge the importance of studying on dynamic streaming type of data. However, our work primarily focuses on the development and evaluation of the LoKO algorithm within well-defined datasets. The type of investigation you mentioned falls under the broader category of Continual Learning [3], which is beyond the primary focus of our paper. Nevertheless, we agree that exploring LoKO in the context of dynamic streaming type of data is an interesting direction for future work, and we appreciate your suggestion.

---

> ### Author Response · Authors · 2024-11-22
> **Covariance Initialization**
>
> As mentioned in response to other reviewers, initializing the covariance matrix requires prior knowledge of the network weights and their associated uncertainties. In the absence of such knowledge, almost any initial values for the covariance matrix can be chosen, provided they are not too close to zero, and in some cases, they are not too far from an upper bound. As long as the initialization falls within an acceptable range, the filter will eventually converge to the optimal values. Notably, the optimal values are not sensitive to the initial covariance matrix. For instance, we present evidence based on the results of CIFAR-100/ViT-B16 for four different covariance initialization values in Figure 7 (Appendix D.3 in the revised paper). As demonstrated, the final outcomes show minimal sensitivity to these initialization.
>
> As discussed in detail in Section 5.3 of the paper; to obtain the upper bounds for our case studies, we conducted a series of experiments across a wide range of initial values and tracked the average online accuracy. If the accuracy doesn’t reach to a specific threshold within a predetermined number of iterations, we consider the test to have diverged. The boundary values derived from this analysis have been reported in Table 4 of the paper, specifically for case studies where an upper bound for the initial values of covariance matrix is defined. We will clarify this point by adding the following discussion to the revised version of the paper:
>
> “Initializing p ̂ requires prior knowledge of the network weights and their associated uncertainties. In the absence of such knowledge, almost any initial values for p ̂ can be chosen, provided they are not too close to zero, and in some cases, they are not too far from an upper bound. Our experiments demonstrate that as long as the initialization falls within an acceptable range, the filter will eventually converge to the optimal values. Notably, the optimal values are not sensitive to the initial covariance matrix.”

---

> ### Author Response · Authors · 2024-11-22
> **Computational Analysis**
>
> Thank you for your constructive feedback. As discussed in Section 4.1, applying a diagonal approximation for the covariance matrix, combined with a low-rank decomposition approach, decrease the computational complexity from quadratic to linear in the number of trainable parameters. To clarify this further, we will include the following explanation in Appendix C.1 of the revised paper:
>
> “Computational Analysis:
> Let n represent the number of parameters, n ̃ the number of trainable parameters (n ̃  ≪n), and m the number of model outputs. The computational complexities of each step in the LoKO algorithm compared to the vanilla Kalman filter are as follows:
> Prediction: The computational complexity of both LoKO and the vanilla Kalman filter for the Prediction step is O(1).
> Pre-Updating: For LoKO, estimating the observation noise covariance using Equation (7) has a complexity of O(m^2). When using Equation (12), the complexity increases to O(m^2 n ̃). In contrast, for the vanilla Kalman filter, the complexity for Equation (7) is O(m^2), while for Equation (12) it is O(m^2 n+n^2 m).
> Updating: The computational complexity for LoKO when calculating the Kalman gain (Equation (11a)) is O(m^3+m^2 n ̃), while for the vanilla Kalman filter (using Equation (3a)), it will be O(m^3+m^2 n+n^2 m). For updating parameters (Equation (11b)), LoKO has a complexity of O(mn ̃), compared to O(mn) for the vanilla Kalman filter. And finally, the complexity for covariance updating (Equation (11c)) in LoKO is O(n ̃), while for the vanilla Kalman filter (using Equation (3c)), it will be O(n^2 m).
> Thus, the total computational complexity in the worst-case scenario, for LoKO is O(m^3+m^2 n ̃), and for the vanilla Kalman filter will be O(m^3+m^2 n+n^2 m). In cases where the number of parameters is significantly larger than output size, the dominant term for LoKO is O(m^2 n ̃), while for the vanilla Kalman filter, it will be O(n^2 m). Therefore, LoKO reduces the computational complexity from quadratic to linear in the number of parameters as well as decreasing the number of trainable parameters (n ̃  ≪n).”
>
> To evaluate the time efficiency of our LoKO algorithm, we compare the number of steps required for convergence, similar to the criterion used in [4]. Convergence is defined as the number of iterations at which the loss stays flat or decreases by less than a specified threshold. The time analysis is presented in Appendix C.2 of the revised paper.

---

> ### Author Response · Authors · 2024-11-22
> **References**
>
> [1]	S.-Y. Liu et al., “DoRA: Weight-Decomposed Low-Rank Adaptation,” Jul. 09, 2024, arXiv: arXiv:2402.09353. Accessed: Nov. 19, 2024. [Online]. Available: http://arxiv.org/abs/2402.09353
>
> [2]	Z. Han, C. Gao, J. Liu, J. Zhang, and S. Q. Zhang, “Parameter-Efficient Fine-Tuning for Large Models: A Comprehensive Survey,” Jul. 12, 2024, arXiv: arXiv:2403.14608. Accessed: Sep. 08, 2024. [Online]. Available: http://arxiv.org/abs/2403.14608
>
> [3]	G. I. Parisi, R. Kemker, J. L. Part, C. Kanan, and S. Wermter, “Continual lifelong learning with neural networks: A review,” Neural Netw., vol. 113, pp. 54–71, May 2019, doi: 10.1016/j.neunet.2019.01.012.
>
> [4]	H. Liu, Z. Li, D. Hall, P. Liang, and T. Ma, “Sophia: A Scalable Stochastic Second-order Optimizer for Language Model Pre-training,” Mar. 05, 2024, arXiv: arXiv:2305.14342. Accessed: Mar. 12, 2024. [Online]. Available: http://arxiv.org/abs/2305.14342

---

> ### Comment · Reviewer_WNJR · 2024-11-25
>
> Thanks. The additional results indeed look promising.
>
> One question regarding to your reply. What do you mean about "weight-decomposed low-rank adaptation technique"? R u referring to the paper "[1] S.-Y. Liu et al., “DoRA: Weight-Decomposed Low-Rank Adaptation,” Jul. 09, 2024, arXiv: arXiv:2402.09353. Accessed: Nov. 19, 2024. [Online]. Available: http://arxiv.org/abs/2402.09353"?

---

> > ### Author Response · Authors · 2024-11-25
> >
> > Thank you for your feedback. Yes, “Weight-Decomposed Low-Rank Adaptation” refers to the new decomposition technique introduced by the cited paper. This method decomposes pre-trained weights into magnitude and direction components to minimize the number of trainable parameters more efficiently.

---

### Official Review · Reviewer_jCMG · 2024-11-02

**Soundness:** 3
**Presentation:** 3
**Contribution:** 3
**Rating:** 6
**Confidence:** 4

**Summary:**

This paper introduces LoKO, a novel optimization algorithm that integrates the Extended Kalman Filter (EKF) with Low-Rank Adaptation (LoRA) for the online fine-tuning of large pre-trained models. By reframing parameter-efficient fine-tuning (PEFT) as a state estimation problem, the authors leverage the EKF to estimate optimal trainable parameters in an online manner. The use of LoRA's low-rank decomposition reduces the number of trainable parameters, and a diagonal approximation of the covariance matrix further decreases computational complexity from quadratic to linear. An Exponential Moving Average (EMA) approach is proposed for estimating the observation noise covariance without additional computational cost. Empirical evaluations on various computer vision and natural language processing tasks demonstrate that LoKO achieves competitive or superior performance compared to traditional gradient-based optimizers like AdamW and AdaGrad.

**Strengths:**

Casting PEFT as a state estimation problem using the Kalman filter is a novel and insightful approach. The combination of LoRA with EKF addresses the scalability issues traditionally associated with Kalman filters in large-scale models. The diagonal approximation of the covariance matrix and EMA-based estimation significantly reduce computational complexity, making the approach practical for large models.

The paper provides a clear and well-founded mathematical formulation of the proposed method. The derivation of the simplified Kalman update equations is thorough, and practical implementation considerations are discussed. The thoughtful discussion on covariance matrix initialization and its impact on performance demonstrates a deep understanding of the method's nuances.

The method is evaluated across multiple domains, including image and text classification tasks, demonstrating its versatility. Thorough comparisons with mainstream optimizers (AdamW and AdaGrad) show that LoKO often achieves better convergence and performance. Detailed analyses on the effects of covariance initialization and the forgetting factor provide valuable insights into the method's behavior.

**Weaknesses:**

The paper lacks a formal convergence analysis of the proposed algorithm. Without theoretical guarantees, it's unclear under what conditions LoKO is expected to perform reliably.
The justification for the diagonal approximation of the covariance matrix is primarily empirical. A theoretical analysis or conditions under which this approximation holds would strengthen the contribution.
The theoretical implications of using EMA for observation noise covariance estimation are not fully explored. Understanding its impact on convergence and stability is important.
The experiments do not include comparisons with other recent PEFT methods such as QLoRA or AdaLoRA. Including these would better position LoKO within the current landscape. Evaluations are conducted over a single epoch, which may not capture the long-term behavior and stability of the optimizer. The focus on online fine-tuning with batch size 1 raises questions about the method's applicability and performance in standard offline settings with larger batch sizes. In some language tasks, LoKO performs slightly worse than AdamW. Investigating these cases could reveal insights into the method's limitations.

While the paper claims reduced computational complexity, it does not provide empirical runtime measurements or resource usage comparisons to substantiate these claims. There is limited discussion on potential trade-offs between computational cost and optimization performance compared to other methods.

**Questions:**

Please address the comments in the weakness part.

---

> ### Author Response · Authors · 2024-11-22
> **General Comment**
>
> We sincerely appreciate your invaluable feedback. We have made every effort to address your concerns and have provided a detailed point-by-point explanation in the following list of responses. Furthermore, all modifications have been clearly highlighted in the revised version of the paper that has been submitted.

---

> ### Author Response · Authors · 2024-11-22
> **Theoretical Analysis**
>
> To clarify, the primary focus of this study is on the empirical investigation of LoKO, rather than on theoretical analysis. There is no formal theoretical proof for LoRA (at least, we could not identify any in the literature). Given the challenges posed by the non-convex loss landscape, relatively few studies have tackled the theoretical analysis of optimizers such as Adam and SGD like [1]. Most works in this area evaluate the effectiveness of the algorithms primarily through empirical evidence. We acknowledge the value of a formal convergence analysis and consider it an interesting direction for future work. However, in the Appendix B, we have provided some theoretical insights as summarized below:
>
> o	Proposition 1: To establish the core contribution of this paper, we proved that the proposed LoKO formulation is mathematically equivalent to the vanilla Kalman filter algorithm when leveraging the low-rank decomposition technique in LoRA and applying a diagonal approximation of the covariance matrix. This proof relies on the properties of the transposed Khatri–Rao product, Einstein notation, and their connections to matrix multiplication.
>
> o	EMA technique for R matrix Estimation: We validated our EMA approach for estimating the noise covariance matrix (R) by employing the zero- and first-order terms of the Taylor series expansion around the most recent parameter updates, and expanding it based on the definition of the noise covariance.

---

> ### Author Response · Authors · 2024-11-22
> **Other PEFT Methods**
>
> Thank you for this valuable suggestion. We find your recommendation highly insightful. We have conducted additional experiments involving the DoRA [2] variant of LoRA, in combination with Kalman, AdamW, and AdaGrad. The results from these experiments are now included in the revised paper, with visualizations in Appendix D.1 (Figure 3, and Figure 4) and further analysis in Table 1, and Table 2, Section 5.2. As shown, the results are promising for the Kalman algorithm, demonstrating its compatibility and effectiveness when integrated with this Weight-Decomposed Low-Rank Adaptation technique.

---

> ### Author Response · Authors · 2024-11-22
> **Single Epoch**
>
> Single epoch fine-tuning is a common practice in scenarios where large pretrained models are adapted to specific tasks or datasets particularly in large-scale datasets, as seen in [3], [4], [5]. However, we have performed multiple epochs of fine-tuning for smaller datasets (COLA and MRPC), as clearly mentioned in Section 5.2 of the paper.

---

> ### Author Response · Authors · 2024-11-22
> **Batch Size**
>
> The primary focus of this paper is on "online fine-tuning" scenarios, and we intentionally emphasize this context. While we acknowledge that offline settings with larger batch sizes are important, they fall outside the scope of this study.

---

> ### Author Response · Authors · 2024-11-22
> **Language Tasks Results**
>
> Thank you for this insightful comment. As discussed in [6], AdamW optimizer benefits from the decoupled weight decay, which has contributed to its strong performance in language modeling tasks. On the other hand, the gradient-free Kalman algorithm used in LoKO does not incorporate a similar regularization mechanism, which may explain the slightly lower performance in certain cases compared to AdamW. We will clarify this point by adding the following discussion to Section 5.2 of the revised version of the paper:
> “As can be seen in Table 2, LoKO demonstrates slightly lower performance than AdamW on certain language tasks. A justification for this observation is the fact that AdamW optimizer benefits from the decoupled weight decay, which has contributed to its strong performance in language modeling tasks [6]. On the other hand, the gradient-free Kalman algorithm used in LoKO does not incorporate a similar regularization mechanism, which may explain the slightly lower performance in certain cases compared to AdamW.”

---

> ### Author Response · Authors · 2024-11-22
> **Computational Analysis**
>
> Thank you for your constructive feedback. As discussed in Section 4.1, applying a diagonal approximation for the covariance matrix, combined with a low-rank decomposition approach, decrease the computational complexity from quadratic to linear in the number of trainable parameters. To clarify this further, we will include the following explanation in the Appendix C.1 of the revised paper:
>
> “Computational Analysis:
> Let n represent the number of parameters, n ̃ the number of trainable parameters (n ̃  ≪n), and m the number of model outputs. The computational complexities of each step in the LoKO algorithm compared to the vanilla Kalman filter are as follows:
> Prediction: The computational complexity of both LoKO and the vanilla Kalman filter for the Prediction step is O(1).
> Pre-Updating: For LoKO, estimating the observation noise covariance using Equation (7) has a complexity of O(m^2). When using Equation (12), the complexity increases to O(m^2 n ̃). In contrast, for the vanilla Kalman filter, the complexity for Equation (7) is O(m^2), while for Equation (12) it is O(m^2 n+n^2 m).
> Updating: The computational complexity for LoKO when calculating the Kalman gain (Equation (11a)) is O(m^3+m^2 n ̃), while for the vanilla Kalman filter (using Equation (3a)), it will be O(m^3+m^2 n+n^2 m). For updating parameters (Equation (11b)), LoKO has a complexity of O(mn ̃), compared to O(mn) for the vanilla Kalman filter. And finally, the complexity for covariance updating (Equation (11c)) in LoKO is O(n ̃), while for the vanilla Kalman filter (using Equation (3c)), it will be O(n^2 m).
> Thus, the total computational complexity in the worst-case scenario, for LoKO is O(m^3+m^2 n ̃), and for the vanilla Kalman filter will be O(m^3+m^2 n+n^2 m). In cases where the number of parameters is significantly larger than output size, the dominant term for LoKO is O(m^2 n ̃), while for the vanilla Kalman filter, it will be O(n^2 m). Therefore, LoKO reduces the computational complexity from quadratic to linear in the number of parameters as well as decreasing the number of trainable parameters (n ̃  ≪n).”
>
> To evaluate the time efficiency of our LoKO algorithm, we compare the number of steps required for convergence, similar to the criterion used in [7]. Convergence is defined as the number of iterations at which the loss stays flat or decreases by less than a specified threshold. The time analysis is presented in Appendix C.2 of the revised paper.

---

> ### Author Response · Authors · 2024-11-22
> **References**
>
> [1]	S. J. Reddi, S. Kale, and S. Kumar, “On the Convergence of Adam and Beyond,” Apr. 19, 2019, arXiv: arXiv:1904.09237. Accessed: Nov. 17, 2024. [Online]. Available: http://arxiv.org/abs/1904.09237
>
> [2]	S.-Y. Liu et al., “DoRA: Weight-Decomposed Low-Rank Adaptation,” Jul. 09, 2024, arXiv: arXiv:2402.09353. Accessed: Nov. 19, 2024. [Online]. Available: http://arxiv.org/abs/2402.09353
>
> [3]	A. Panda, B. Isik, X. Qi, S. Koyejo, T. Weissman, and P. Mittal, “Lottery Ticket Adaptation: Mitigating Destructive Interference in LLMs,” Jun. 25, 2024, arXiv: arXiv:2406.16797. Accessed: Nov. 17, 2024. [Online]. Available: http://arxiv.org/abs/2406.16797
>
> [4]	A. Komatsuzaki, “One Epoch Is All You Need,” Jun. 16, 2019, arXiv: arXiv:1906.06669. Accessed: Nov. 17, 2024. [Online]. Available: http://arxiv.org/abs/1906.06669
>
> [5]	S. Chen, M. Jiang, and Q. Zhao, “What Do Deep Saliency Models Learn about Visual Attention?,” Oct. 14, 2023, arXiv: arXiv:2310.09679. Accessed: Nov. 17, 2024. [Online]. Available: http://arxiv.org/abs/2310.09679
>
> [6]	S. Xie and Z. Li, “Implicit Bias of AdamW: $\ell_∞$-Norm Constrained Optimization,” in Proceedings of the 41st International Conference on Machine Learning, PMLR, Jul. 2024, pp. 54488–54510. Accessed: Nov. 17, 2024. [Online]. Available: https://proceedings.mlr.press/v235/xie24e.html
>
> [7]	H. Liu, Z. Li, D. Hall, P. Liang, and T. Ma, “Sophia: A Scalable Stochastic Second-order Optimizer for Language Model Pre-training,” Mar. 05, 2024, arXiv: arXiv:2305.14342. Accessed: Mar. 12, 2024. [Online]. Available: http://arxiv.org/abs/2305.14342

---

### Official Review · Reviewer_AMtv · 2024-11-04

**Soundness:** 3
**Presentation:** 2
**Contribution:** 2
**Rating:** 3
**Confidence:** 3

**Summary:**

Low-Rank Kalman Optimizer (LoKO) is a novel optimizer for Parameter-Efficient Fine-Tuning (PEFT) aimed specifically at LoRA. LoKO frames fine-tuning as an optimal filtering problem, where trainable parameters are estimated as state variables updated dynamically with new observations. By leveraging Kalman’s recursive estimation, LoKO achieves efficient convergence with fewer updates. Additionally, the optimizer incorporates a diagonal approximation for the covariance matrix, which reduces computational complexity from quadratic to linear in the number of parameters.

**Strengths:**

LoKO shows that Kalman optimizers work similarly well with LoRA as they do with bigger weight matrices in a neural network with better and faster convergence and efficient online learning.

**Weaknesses:**

"Novelty" The novelty of the work seems hard to gauge and should be explicitly noted in the paper. The use of EKF as an alternative optimizing strategy has already been pursued and its efficacy for faster convergence has been demonstrated in previous works some of which are cited in the paper. The diagonal approximation (amongst other approximation attempts as in the cited, Chang et al.) for the covariance matrix to reduce computations has also been utilized in a number of previous works regarding Kalman filtering. The novelty in this work seems to be in the application to the finetuning of neural networks using a specific low rank adapter, LoRA. There are no new findings or theoretical insights proposed other than what is expected of training a subset of neural weights (LoRA) with a Kalman optimizer. Unfortunately, this does not seem to be adequate especially given the limited diversity of experiments (explained below) and applicability to a single type of low rank adapter.

The gaussian assumption for the neural parameters and noise are unsupported by any empirical results in the paper.

"Diagonal Approximation" LoKO relies on a diagonal approximation of the covariance matrix to reduce computational costs. While effective, this approximation limits the optimizer's ability to capture parameter correlations fully, which could lead to suboptimal updates for tasks with high-dimensional dependencies​. Experiments on complex tasks like text-to-image generation or others should be utilized to ablate this aspect of the method.

In addition, the sensitivity to initialization of the covariance matrix requires significant ablation across different architectures and tasks to investigate its failings and properties, and whether it varies across different modalities.

"Noise Approximation" The exponential moving average (EMA) approach for noise covariance estimation may be unstable, particularly when new data distributions diverge significantly from the initial training data. It is necessary to test LoKO in OOD generalization tasks to ablate this approximation strategy.

Given this work relates to optimization specific to PeFT methods, its necessary to show its efficacy on different low rank adapters to show similar results and properties can be observed from them. Comparisons with DoRA, X-LoRA, etc. would be great additions to the work.

**Questions:**

- What is the novelty of the work beyond application to LoRA?
- Is this applicable beyond LoRA to other methods?
- Does the diagonal approximation cause problems in harder tasks like text-to-image generation?
- Is OOD generalization adversely affected due to the noise approximation?

---

> ### Author Response · Authors · 2024-11-22
> **General Comment**
>
> First of all, we sincerely thank you for the invaluable feedback. We have carefully considered all your comments and made every effort to address your concerns comprehensively. All modifications have been clearly highlighted in the revised version of the paper that has been submitted. Below, we provide a detailed, point-by-point response to each of your comments:

---

> ### Author Response · Authors · 2024-11-22
> **Novelty**
>
> To clarify, the novelty is explicitly addressed in the paper at the sections of Introduction and Related Works, which is again summarized below:
>
> o	We employ the Low-Rank Adaptation (LoRA) method, introduced by Hu et al. (2021), to significantly reduce the number of trainable parameters, and scale up the Kalman algorithm, enhancing its compatibility for fine-tuning of large models.
>
> o	We adopt a diagonal approximation for the covariance matrix P to reduce the computational overhead from quadratic to linear and take advantage of the exponential moving average (EMA) for estimating the matrix R to improve performance without additional computational cost.
>
> o	We conduct various experiments to demonstrate LoKO’s performance in online fine-tuning tasks across computer vision and language modeling domains.
>
> o	To the best of our knowledge, this is the first successful attempt to fine-tune large models, including transformers with millions of parameters and complex architectures, using the Kalman filter algorithm.
>
> We would like to emphasize that all previous related works and their relation to our work have been clearly explained and cited, including works with diagonal approximation: [1], [2], and [3] that is also mentioned in your comment. If there are any related or similar works that highlight areas of lower novelty in our research, we would greatly appreciate it if the reviewer could kindly provide specific references.

---

> > ### Comment · Reviewer_AMtv · 2024-11-24
> > **Novelty queries**
> >
> > Thank you for the reply.
> >
> > Unfortunately, the novelty is again not clear enough here.
> > You mention that you employ LoRA (only LoRA, importantly) "to significantly reduce the number of trainable parameters", and then mention "scale up the Kalman algorithm". If you are using Kalman algorithm for optimizing only LoRA parameters - that is not scaling up, is it? The point of LoRA is to drastically reduce the number of trainable parameters, often at the cost of finetuning performance - there is no scaling up there. Please clarify this point.
> >
> > Is the adoption of the diagonal approximation a novelty? Has no work done this in the context of Kalman based optimization before?
> >
> > Although I'd hate to ask you to do more experiments, but unfortunately, given your claims in the work regarding a new optimization process for a *specific* low rank adapter, doing experiments only on image classification and a small subset of the GLUE benchmark is not adequate evidence for a paper which is largely backed by its results. Given the broad application of LoRA across a huge number of domains and modalities, there should be more such experiments, even at a smaller scale, to show that your optimization method works all these different scenarios. Right now, it is very hard to gauge whether the method is beneficial or not.
> >
> > I'd agree that is an application (probably novel in its application, although I am not sure) of Kalman based optimization to finetune LoRAs and not entire large models, however I'm not sure how this is any different from applying same optimization to other neural models.

---

> > > ### Author Response · Authors · 2024-11-25
> > > **Novelty Response**
> > >
> > > Thank you for your response. We acknowledge that our work is based on LoRA as a state-of-the-art parameter-efficient fine-tuning. We claim that the feasibility and performance of combining the Kalman filter with the LoRA technique in online fine-tuning have not yet been fully explored. To the best of our knowledge, no prior study has investigated this specific combination. If such studies exist, we would greatly appreciate it if you could kindly provide specific references.
> > > The exceptional performance of the Kalman filter in “online learning”, as demonstrated in prior works cited in our paper, motivated us to extend its application to “online fine-tuning” in combination with LoRA. The novelty of our work lies in evaluating the feasibility of using the Kalman filter to update low-rank decomposed matrices, whereas previous Kalman-based works focused on training full weight matrices “W”. LoRA restricts updates to two low-rank matrices, “A” and “B”, which collectively approximate the original weight matrix through their product (BA). The initialization of these matrices, along with LoRA's specific design, could impact the compatibility and performance of the Kalman filter. Our contribution demonstrated that the Kalman filter can be effectively adapted to this context, and achieve competitive performance in online fine-tuning classification tasks. Moreover, to the best of our knowledge, this is the first successful attempt to fine-tune complex models like transformers using the Kalman filter algorithm.
> > >
> > > Regarding the statement of “scaling up Kalman algorithm”, thank you for mentioning this point. We have revised the paper to ensure accuracy and have removed this statement accordingly.
> > >
> > > Regarding the diagonal approximation: While we recognize prior works that employ diagonal approximations (as cited in the paper), we believe there is no direct similarity in this area. We acknowledge that the adoption of a diagonal approximation for the covariance matrix in Kalman-based optimization is not our primary contribution. However, to the best of our knowledge, this is the first study to integrate such an approximation into the context of fine-tuning large-scale models with LoRA, alongside the EMA technique for noise covariance estimation.
> > >
> > > Regarding experiments, our experiments already span multiple domains on some representative benchmarks using models large enough. The results have demonstrated the feasibility and effectiveness of our approach. As a proof-of-concept, we believe an exhaustive validation across all benchmarks is unnecessary, given the associated costs in time and resources. Furthermore, unlike LoRA, which primarily focuses on language datasets, we have conducted a variety of experiments and further extended our approach to computer vision datasets, which demonstrates the effectiveness of our method beyond language datasets.
> > >
> > > Finally, to address your concern more effectively and clarify our contribution more accurately, we revise the entire paper, particularly the Introduction section, and present our contributions as outlined below:
> > > “• Based on the Low-Rank Adaptation (LoRA) method, introduced by Hu et al. (2021), which significantly reduces the number of trainable parameters, we demonstrated its compatibility with the Kalman filter. Also, we showed that this combination offers faster performance than traditional optimizers in online fine-tuning scenarios.
> > > • We employ a diagonal approximation for the covariance matrix P, a common approach to reduce the computational overhead from quadratic to linear. By integrating this with the exponential moving average (EMA) for estimating the matrix R and incorporating it into the LoRA framework, we achieve improved performance without additional computational cost.
> > > • We conduct various experiments to demonstrate LoKO’s performance in online fine-tuning classification tasks across computer vision and language modeling domains.
> > > • To the best of our knowledge, this is the first successful attempt to fine-tune large and complex models, including transformers with millions of parameters, using the Kalman filter algorithm.”

---

> ### Author Response · Authors · 2024-11-22
> **Gaussian Assumption**
>
> We would like to clarify that the Gaussian assumption applies to each individual neural parameter, representing the uncertainty associated with each parameter, as discussed in [4]. It does not imply a single scalar Gaussian distribution over “all” neural parameters.

---

> > ### Comment · Reviewer_AMtv · 2024-11-24
> > **Gaussian Assumption**
> >
> > Can you please clarify why the gaussian assumption in [4] is applicable to neural network parameters when [4] does not consider them?

---

> > > ### Author Response · Authors · 2024-11-25
> > > **Gaussian Assumption Response**
> > >
> > > The Gaussian assumption for representing the uncertainty of individual states is a fundamental assumption of the Kalman filter, as outlined in [1]. In the context of neural networks, when the Kalman filter is used as an optimizer, it is widely accepted in the literature that the uncertainty of individual neural parameters follows a Gaussian distribution, as demonstrated in [2], [3, p. 202], [4], [5], [6], [7], [8], [9], [10].
> > > Regarding the Gaussian assumption of the observation noise, as discussed in Section 4.1 of our paper, we have extended our work to handle non-Gaussian observation noise by considering a broader class of distributions from the exponential family. This technique is detailed in [4].
> > >
> > > References:
> > >
> > > [1]	G. Welch, “An Introduction to the Kalman Filter,” 1997.
> > >
> > > [2]	P. G. Chang, G. Durán-Martín, A. Y. Shestopaloff, M. Jones, and K. Murphy, “Low-rank extended Kalman filtering for online learning of neural networks from streaming data,” Jun. 27, 2023, arXiv: arXiv:2305.19535. Accessed: May 31, 2024. [Online]. Available: http://arxiv.org/abs/2305.19535
> > >
> > > [3]	P. G. Chang, M. Jones, and K. Murphy, “On diagonal approximations to the extended Kalman filter for online training of Bayesian neural networks”.
> > >
> > > [4]	Y. Ollivier, “Online Natural Gradient as a Kalman Filter,” 2017, doi: 10.48550/ARXIV.1703.00209.
> > >
> > > [5]	S. Murtuza and S. F. Chorian, “Node decoupled extended Kalman filter based learning algorithm for neural networks,” in Proceedings of 1994 9th IEEE International Symposium on Intelligent Control, Aug. 1994, pp. 364–369. doi: 10.1109/ISIC.1994.367790.
> > >
> > > [6]	G. V. Puskorius and L. A. Feldkamp, “Decoupled extended Kalman filter training of feedforward layered networks,” in IJCNN-91-Seattle International Joint Conference on Neural Networks, Seattle, WA, USA: IEEE, 1991, pp. 771–777. doi: 10.1109/IJCNN.1991.155276.
> > >
> > > [7]	S. Shah, F. Palmieri, and M. Datum, “Optimal filtering algorithms for fast learning in feedforward neural networks,” Neural Netw., vol. 5, no. 5, pp. 779–787, Sep. 1992, doi: 10.1016/S0893-6080(05)80139-X.
> > >
> > > [8]	S. Shah and F. Palmieri, “MEKA-a fast, local algorithm for training feedforward neural networks,” in 1990 IJCNN International Joint Conference on Neural Networks, San Diego, CA, USA: IEEE, 1990, pp. 41–46 vol.3. doi: 10.1109/IJCNN.1990.137822.
> > >
> > > [9]	D. W. Ruck, S. K. Rogers, M. Kabrisky, P. S. Maybeck, and M. E. Oxley, “Comparative analysis of backpropagation and the extended Kalman filter for training multilayer perceptrons,” IEEE Trans. Pattern Anal. Mach. Intell., vol. 14, no. 6, pp. 686–691, Jun. 1992, doi: 10.1109/34.141559.
> > >
> > > [10]	S. Singhal and L. Wu, “Training Multilayer Perceptrons with the Extended Kalman Algorithm,” in Advances in Neural Information Processing Systems, Morgan-Kaufmann, 1988. Accessed: Jan. 12, 2024. [Online]. Available: https://proceedings.neurips.cc/paper/1988/hash/38b3eff8baf56627478ec76a704e9b52-Abstract.html

---

> ### Author Response · Authors · 2024-11-22
> **Diagonal Approximation**
>
> The adoption of a diagonal approximation for large-scale covariance and Hessian matrices is a widely accepted practice in machine learning due to its computational efficiency. For instance, [5] utilizes a diagonal Hessian approximation in the development of a second-order optimizer. Or, [6] demonstrates that the Adam algorithm effectively employs a diagonal approximation of the pure Newton’s method update. Furthermore, it is well-known that even using a full covariance matrix does not guarantee convergence to a global optimum, given the highly non-convex nature of the loss landscape. Lastly, we note that text-to-image generation, while an important area of research, falls outside the scope of this paper.

---

> > ### Comment · Reviewer_AMtv · 2024-11-24
> > **Diagonal Approximation query**
> >
> > Thank for the citations. However, my point still stands. If the author means that their method is only applicable to image classification and subset of GLUE task, please state that in the paper.

---

> > > ### Author Response · Authors · 2024-11-25
> > > **Diagonal Approximation Response**
> > >
> > > Thank you for this comment. We have previously mentioned this in the Conclusion section. Now we have revised the explanation for greater clarity. The updated text in the Conclusion section will read as follows:
> > > “Although this paper focuses on online fine-tuning in computer vision and language models for classification tasks, further evaluations are necessary across a broader range of domains, e.g., reinforcement learning. We leave these explorations for future work.”

---

> ### Author Response · Authors · 2024-11-22
> **Covariance Initialization**
>
> As mentioned in response to other reviewers, initializing the covariance matrix requires prior knowledge of the network weights and their associated uncertainties. In the absence of such knowledge, almost any initial values for the covariance matrix can be chosen, provided they are not too close to zero, and in some cases, they are not too far from an upper bound. As long as the initialization falls within an acceptable range, the filter will eventually converge to the optimal values. Notably, the optimal values are not sensitive to the initial covariance matrix. For instance, we present evidence based on the results of CIFAR-100/ViT-B16 for four different covariance initialization values in Figure 7 (Appendix D.3 in the revised paper). As demonstrated, the final outcomes show minimal sensitivity to the initialization.
>
> As discussed in detail in Section 5.3 of the paper; to obtain the upper bounds for our case studies, we conducted a series of experiments across a wide range of initial values and tracked the average online accuracy. If the accuracy doesn’t reach to a specific threshold within a predetermined number of iterations, we consider the test to have diverged. The boundary values derived from this analysis have been reported in Table 4 of the paper, specifically for case studies where an upper bound for the initial values of covariance matrix is defined. We will clarify this point by adding the following discussion to the revised version of the paper:
> “Initializing p ̂ requires prior knowledge of the network weights and their associated uncertainties. In the absence of such knowledge, almost any initial values for p ̂ can be chosen, provided they are not too close to zero, and in some cases, they are not too far from an upper bound. Our experiments demonstrate that as long as the initialization falls within an acceptable range, the filter will eventually converge to the optimal values. Notably, the optimal values are not sensitive to the initial covariance matrix.”

---

> ### Author Response · Authors · 2024-11-22
> **Noise Approximation**
>
> We appreciate your comment and acknowledge the importance of studying OOD generalization. However, our work primarily focuses on the development and evaluation of the LoKO algorithm within the Independent and Identically Distributed dataset. OOD generalization typically requires addressing issues in distributional shift scenarios [7], which lies outside the primary focus of our current work. Although we recognize that EMA-based noise covariance estimation may be sensitive to distribution shifts, we believe that the current methodology is appropriate for the setting and domain considered in our experiments. To address your concern more effectively, it would be helpful if you could clarify the specific types of OOD tasks you have in mind.

---

> > ### Comment · Reviewer_AMtv · 2024-11-24
> > **Noise Approximation**
> >
> > The point I'm trying to make is that this noise approximation makes the method more susceptible to poorer generalization capabilities as compared to simple the conventional optimization? Intuitively, it appears so to me and could be an drawback of the LoKO method. Ablating this would be important.

---

> > > ### Author Response · Authors · 2024-11-25
> > > **Noise Approximation Response**
> > >
> > > To evaluate the sensitivity of LoKO to OOD data, we conducted experiments on MNIST classification task in an online manner. To generate the out-of-distributed MNIST images, we applied a combination of random rotation (30 degrees) and color jitter (with a brightness and contrast adjustment of 0.5). Then, we inserted these OOD images into the online fine-tuning process by including one OOD sample after every 100 normal samples. The results have been added to Appendix D.6 of the revised paper. As shown in Figure 9, both LoKO and AdamW exhibit some sensitivity to OOD data due to their use of the EMA technique. However, the results for LoKO demonstrate that it adapts more quickly to the shifted distribution compared to AdamW. In contrast, AdaGrad shows less sensitivity to OOD data, as it does not incorporate the EMA technique for the first moment.

---

> ### Author Response · Authors · 2024-11-22
> **Other variants of LoRA**
>
> Thank you for this valuable suggestion. We find your recommendation highly insightful. We have conducted additional experiments involving the DoRA [8] variant of LoRA, in combination with Kalman, AdamW, and AdaGrad. The results from these experiments are now included in the revised paper, with visualizations in Appendix D.1 (Figure 3, and Figure 4) and further analysis in Table 1, and Table 2, Section 5.2. As shown, the results are promising for the Kalman algorithm, demonstrating its compatibility and effectiveness when integrated with this Weight-Decomposed Low-Rank Adaptation technique.

---

> ### Author Response · Authors · 2024-11-22
> **References**
>
> [1]	S. Murtuza and S. F. Chorian, “Node decoupled extended Kalman filter based learning algorithm for neural networks,” in Proceedings of 1994 9th IEEE International Symposium on Intelligent Control, Aug. 1994, pp. 364–369. doi: 10.1109/ISIC.1994.367790.
>
> [2]	P. G. Chang, M. Jones, and K. Murphy, “On diagonal approximations to the extended Kalman filter for online training of Bayesian neural networks”.
>
> [3]	P. G. Chang, G. Durán-Martín, A. Y. Shestopaloff, M. Jones, and K. Murphy, “Low-rank extended Kalman filtering for online learning of neural networks from streaming data,” Jun. 27, 2023, arXiv: arXiv:2305.19535. Accessed: May 31, 2024. [Online]. Available: http://arxiv.org/abs/2305.19535
>
> [4]	G. Welch, “An Introduction to the Kalman Filter,” 1997.
>
> [5]	H. Liu, Z. Li, D. Hall, P. Liang, and T. Ma, “Sophia: A Scalable Stochastic Second-order Optimizer for Language Model Pre-training,” Mar. 05, 2024, arXiv: arXiv:2305.14342. Accessed: Mar. 12, 2024. [Online]. Available: http://arxiv.org/abs/2305.14342
>
> [6]	I. Molybog et al., “A Theory on Adam Instability in Large-Scale Machine Learning,” Apr. 25, 2023, arXiv: arXiv:2304.09871. Accessed: Nov. 16, 2024. [Online]. Available: http://arxiv.org/abs/2304.09871
>
> [7]	J. Liu et al., “Towards Out-Of-Distribution Generalization: A Survey,” Jul. 27, 2023, arXiv: arXiv:2108.13624. doi: 10.48550/arXiv.2108.13624.
>
> [8]	S.-Y. Liu et al., “DoRA: Weight-Decomposed Low-Rank Adaptation,” Jul. 09, 2024, arXiv: arXiv:2402.09353. Accessed: Nov. 19, 2024. [Online]. Available: http://arxiv.org/abs/2402.09353

---

### Meta-Review · Area_Chair_yZwu · 2024-12-23

**Metareview:**

## Summary

LoKO is a novel optimization algorithm that integrates the Extended Kalman Filter (EKF) with Low-Rank Adaptation (LoRA) for online fine-tuning of large pre-trained models. It uses LoRA's low-rank decomposition to reduce the number of trainable parameters and a diagonal approximation of the covariance matrix, reducing computational complexity. LoKO achieves competitive or superior performance compared to traditional gradient-based optimizers, showing promising results on computer vision and language modeling benchmarks.


## Strengths
* LoKO combines Kalman optimizers with LoRA for efficient online learning and faster convergence, whereas it addresses scalability issues in large-scale models.
* LoKO is evaluated across multiple domains on mainstream optimizers, including image and text classification tasks, and often achieves better convergence and performance.
* The method achieves performance comparable to existing gradient descent-based algorithms across multiple models and datasets.

## Weaknesses
* The diagonal approximation for the covariance matrix is used to reduce computational costs, but it limits the optimizer's ability to capture parameter correlations fully.
* The paper does not provide a formal convergence analysis of the proposed algorithm, and the theoretical implications of using EMA for observation noise covariance estimation are not fully explored.
* The paper does not include comparisons with other recent PEFT methods such as QLoRA or AdaLoRA, and lacks a comparison of the runtime efficiency between LoKO and the baseline methods.
* The focus on online fine-tuning with batch size 1 raises questions about the method's applicability and performance in standard offline settings with larger batch sizes.
* The paper claims reduced computational complexity but does not provide empirical runtime measurements or resource usage comparisons to substantiate these claims.
* The LoKO method assumes that the model's trainable parameters remain unchanged during the observation process, which is often difficult to satisfy in online scenarios.

## Conclusions
Based on the reviews, author feedback and rebuttals, the paper has been improved thanks to the effort of the authors by adding more experiments, and clarification of the text. However, there were some concerns that did not satisfy the reviewers. For instance, there are a few citations about low-rank kalman filtering like *G. Chang, G. Durán-Martín, A. Y. Shestopaloff, M. Jones, and K. Murphy, “Low-rank extended Kalman filtering for online learning of neural networks from streaming data,” Jun. 27, 2023* and have a clear differentiation instead of just citing them briefly. Therefore, the paper should be improved before being accepted.

**Additional Comments On Reviewer Discussion:**

The author created an extensive feedback addressing all the concerns and introducing them in the final manuscript. There was a long discussion about a few points that were partially satisfied. Moreover, a better differentiation of the proposed approach should be addressed before accepting the paper.

---

### Decision · Program_Chairs · 2025-01-22

Reject